# Principal cells of the brainstem's interaural sound level detector are temporal differentiators rather than integrators

Tom P Franken[1†], Philip X Joris[1]*, Philip H Smith[2]

[1]Laboratory of Auditory Neurophysiology, KU Leuven, Leuven, Belgium; [2]University of Wisconsin, Madison, United States

**Abstract** The brainstem's lateral superior olive (LSO) is thought to be crucial for localizing high-frequency sounds by coding interaural sound level differences (ILD). Its neurons weigh contralateral inhibition against ipsilateral excitation, making their firing rate a function of the azimuthal position of a sound source. Since the very first in vivo recordings, LSO principal neurons have been reported to give sustained and temporally integrating 'chopper' responses to sustained sounds. Neurons with transient responses were observed but largely ignored and even considered a sign of pathology. Using the Mongolian gerbil as a model system, we have obtained the first in vivo patch clamp recordings from labeled LSO neurons and find that principal LSO neurons, the most numerous projection neurons of this nucleus, only respond at sound onset and show fast membrane features suggesting an importance for timing. These results provide a new framework to interpret previously puzzling features of this circuit.

DOI: https://doi.org/10.7554/eLife.33854.001

*For correspondence:
Philip.Joris@med.kuleuven.be

Present address: †Systems Neurobiology Laboratories, The Salk Institute for Biological Studies, La Jolla, United States

Competing interests: The authors declare that no competing interests exist.

## Introduction

The lateral superior olive (LSO) in the superior olivary complex is a major component of the brainstem high-frequency sound localization circuitry. Its neurons are sensitive to sound level differences between the ears (interaural level differences or ILD; see *Tollin, 2003*, for review). Neurons in the LSO accomplish this by weighing the sound input to the ipsilateral ear, conveyed via an excitatory (glutamatergic) projection from the ipsilateral cochlear nucleus (*Cant, 1991*; *Smith et al., 1993*; *Doucet and Ryugo, 2003*), against the sound input to the contralateral ear, conveyed via an inhibitory (glycinergic) afferent from the homolateral medial nucleus of the trapezoid body (*Figure 1A*) (*Glendenning et al., 1985*; *Banks and Smith, 1992*; *Sommer et al., 1993*). This process results in firing rates that vary monotonically with ILD, and this tuning is conveyed to targets in the midbrain: the ipsilateral and contralateral inferior colliculi (IC) and the contralateral dorsal nucleus of the lateral lemniscus (DNLL) (*Glendenning and Masterton, 1983*; *Moore, 1988*; *Saint Marie et al., 1989*; *Glendenning et al., 1992*). Reports based on extracellular recordings have consistently indicated that the majority of LSO neurons generate sustained 'chopper' responses to ipsilateral tones and thus temporally integrate auditory input (e.g. *Guinan et al., 1972a*, *1972b*; *Tsuchitani, 1977*; *Greene and Davis, 2012*). Typically, this feature is used as a physiological criterion to define LSO units (e.g. *Finlayson and Caspary, 1989*; *Joris and Yin, 1995*; *Tollin et al., 2008*; *Tsai et al., 2010*).

Morphologically, at least 5 types of LSO neurons have been defined in cat and gerbil, based on dendritic configuration and the size, location and synaptic coverage of the cell body (*Helfert and Schwartz, 1986*; *1987*; *Rietzel and Friauf, 1998*). It is unclear whether the chopper response pattern generalizes across these morphological cell types. Seventy to eighty percent of LSO cells have a

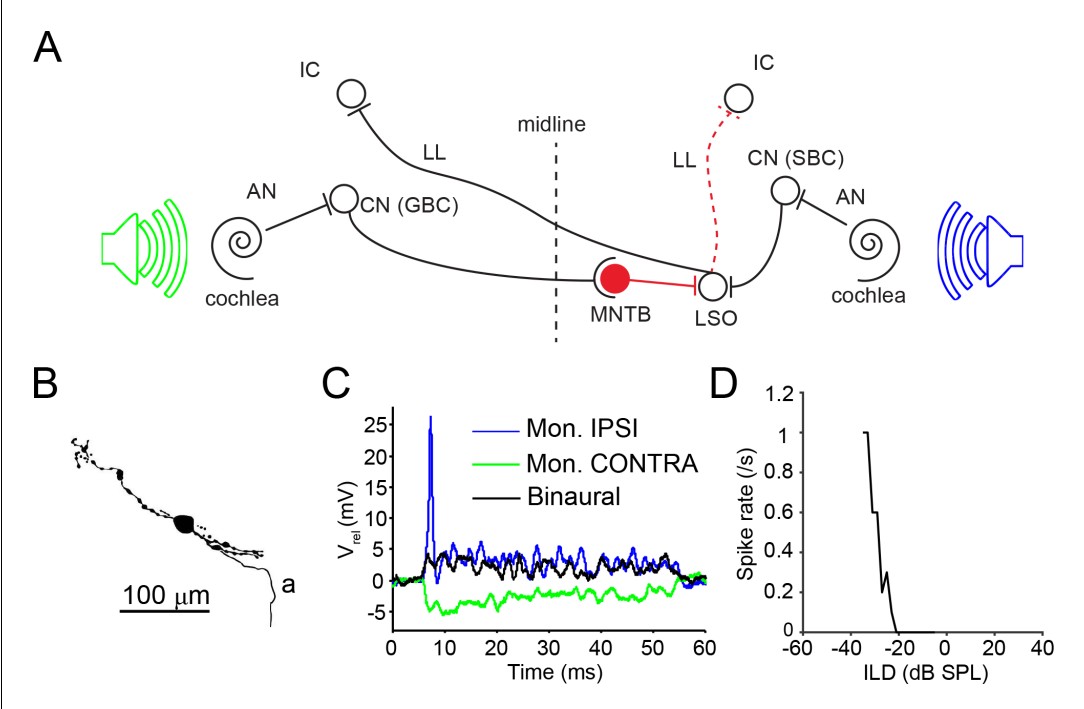

**Figure 1.** LSO circuit and basic properties. (**A**) Outline of the major nuclei/cells involved in the LSO circuit. AN: auditory nerve; CN: cochlear nucleus; GBC: globular bushy cell; IC: inferior colliculus; MNTB: medial nucleus of the trapezoid body; LL: lateral lemniscus; LSO: lateral superior olive; SBC: spherical bushy cell. Speakers symbolize ipsilateral (blue) and contralateral (green) auditory stimulation. (**B**) Camera lucida reconstruction for a neuron with CF = 12 kHz. a: axon. (**C**) Example monaural ipsilateral (blue), monaural contralateral (green) and binaural (black) responses to short tones (12 kHz, 70 dB) for the neuron shown in panel B. (**D**) ILD function for the neuron in panel B (12 kHz, ipsilateral sound level fixed at 85 dB).

DOI: https://doi.org/10.7554/eLife.33854.002

bipolar dendritic configuration but this group can be further subdivided at the electron microscopic level based on the synaptic coverage of the cell body. The majority can be classified as principal cells with cell bodies displaying an extensive synaptic coverage, but the cell bodies of a smaller percentage of bipolar cells are sparsely innervated and termed class 5 cells. Around 13% of the LSO cell population are classified as multiplanar cells, with multipolar dendritic trees and an extensive synaptic coverage of the cell body, or marginal cells, with multipolar or unipolar dendritic trees and sparsely innervated cell bodies that are situated along the margins of the body of the LSO neuropil. The remaining LSO cells are classified as small, have very sparsely innervated cell bodies, and are thought to belong to the olivocochlear system (*Radtke-Schuller et al., 2015*). Presently there is no evidence as to whether any auditory response features differ for these subtypes. A retrograde labeling study indicated that most LSO neurons that project to the inferior colliculus are principal cells (*Saint Marie et al., 1989*). Implicitly, it has been assumed that the most frequently observed response pattern, that is a sustained response with a chopping pattern, is associated with principal cells.

Intracellular evaluation of the synaptic and intrinsic membrane features of LSO cells has been the subject of a number of in vitro slice studies (*Sanes, 1990*; *Wu and Kelly, 1994*; *Adam et al., 1999*; *Barnes-Davies et al., 2004*; *Sterenborg et al., 2010*; *Walcher et al., 2011*; *Jurkovičová-Tarabová et al., 2012*; *Alamilla and Gillespie, 2013*; *Kotak and Sanes, 2014*; *Remme et al., 2014*). In these studies as well, there is evidence for multiple response patterns in response to current injection, and debate regarding the meaning of these patterns and reasons for discrepancies between studies. Non-olivocochlear LSO neurons were associated with both onset and sustained firing to depolarizing current, where the onset response seems to depend on the higher concentration of a low voltage-activated potassium channel (Kv1). In almost all of these studies, cells with these onset and sustained firing properties were classified as 'principal' typically based on visualization of the cell body size and shape, and the presence of I_h current. Only rarely (*Adam et al., 1999*; *Barnes-*

*Davies et al., 2004*) was labeling done to verify the dendritic tree configuration, and the synaptic coverage of the cell body was never evaluated.

Only one previously published report (*Finlayson and Caspary, 1989*) describes in vivo intracellular responses of cells localized to the LSO, in chinchilla. As with many of the extracellular reports, this paper only describes cells with chopper responses to ipsilateral tones. Two cells were labeled with Lucifer Yellow and classified as principal cells based on dendritic configuration but no electron microscopy was done. In almost all of the intracellular recordings reported in that study, ipsilateral acoustic stimuli at the cell's characteristic frequency (CF: frequency of lowest threshold) generated 'robust' suprathreshold excitatory postsynaptic potentials (EPSPs) while contralateral stimuli generated inhibitory postsynaptic potentials (IPSPs). In the few cases studied with binaural stimulation, the inhibitory input was capable of suppressing the spike output generated by the ipsilaterally-evoked EPSPs.

Using patch electrodes we have intracellularly recorded from cells in the gerbil LSO in vivo and have biocytin-labeled examples of the four anatomical non-olivocochlear cell types in this species. We find that the temporal integrative type of 'chopper' response classically associated with this nucleus is not the response generated by its principal cells. Rather, both the response to current injection and to sound indicate that these cells temporally differentiate rather than integrate.

## Results

We have obtained in vivo whole-cell recordings from 25 neurons in the gerbil LSO, of which 16 were labeled with biocytin (*Supplementary file 1*; example cell camera lucida reconstruction in *Figure 1B*). For all cells, ipsilateral tones generated EPSPs and often action potentials (*Figure 1C*, blue trace). In response to contralateral tones, IPSPs were elicited (*Figure 1C*, green trace) in all cases tested (including all unlabeled neurons). When sound was presented binaurally, the ipsilateral depolarization and ensuing spiking could be suppressed by the contralateral inhibition (*Figure 1C*, black trace), generating sensitivity to ILD (*Figure 1D*) in all cases tested (including all unlabeled neurons). In a further 11 LSO neurons, spike output was studied without gaining intracellular access.

### Identification of principal cells and non-principal cells

Light and electron microscopic analysis of labeled neurons allowed us to unambiguously identify five neurons as principal cells (*Figure 2A*, green symbols; *Supplementary file 1*, cells 1–5), using previously reported features for gerbil LSO neurons (*Helfert and Schwartz, 1987*). At the light microscopic level, principal cells have cell bodies located centrally in the LSO and bipolar dendritic arbors in the transverse plane (*Figure 2A*, top row). At the electron microscopic (E.M.) level, principal cells are characterized by high levels of cell body synaptic coverage (*Figure 3*).

In addition, we identified seven non-principal LSO cells, which can be further subdivided in three groups (*Supplementary file 1*, cells 10–16). We identified one class 5 cell, which can only be distinguished from principal cells at the E.M. level by low cell body synaptic coverage. Five other cells could be classified as marginal cells - these cell bodies were located in the LSO marginal area (*Figure 2A*, bottom row). The dendritic trees of two of these (second and fourth from left, cells 12 and 13) were not bipolar in contrast to LSO principal/class 5 cells. One (leftmost, cell 11) had a bipolar dendritic configuration but the orientation of this tree was not tilted at the angle seen for principal/class 5 cells at this location. The remaining cell (third from left, cell 14) had a bipolar morphology similar to principal/class 5 cells but the dendrites heading ventrally projected outside the boundaries of the LSO. E.M. analysis of 4 marginal cell bodies (*Figure 3*) confirmed that they had synaptic coverage levels that fell well below those for gerbil principal cells. Finally, we identified one multiplanar cell, whose cell body was located in the body of the LSO and had a multipolar dendritic tree (*Figure 2A*, near figure center, cell 15). This cell had an extensive synaptic coverage of 75.3% that was consistent with the range of coverages reported by Helfert and Schwartz (*Helfert and Schwartz, 1987*) for this cell type (*Figure 3*). None of our labeled cells had light and electron microscopic features corresponding to 'small' cells in gerbil LSO, most or all of which are thought to be part of the olivocochlear system (*Radtke-Schuller et al., 2015*).

We were able to determine the frequency tuning to ipsilateral stimuli for the labeled cells and measured CF or best frequency (BF, frequency eliciting the largest number of spikes or EPSPs). CF or BF shows a clear increase to higher values towards the medial pole of the nucleus (*Figure 2A/B*),

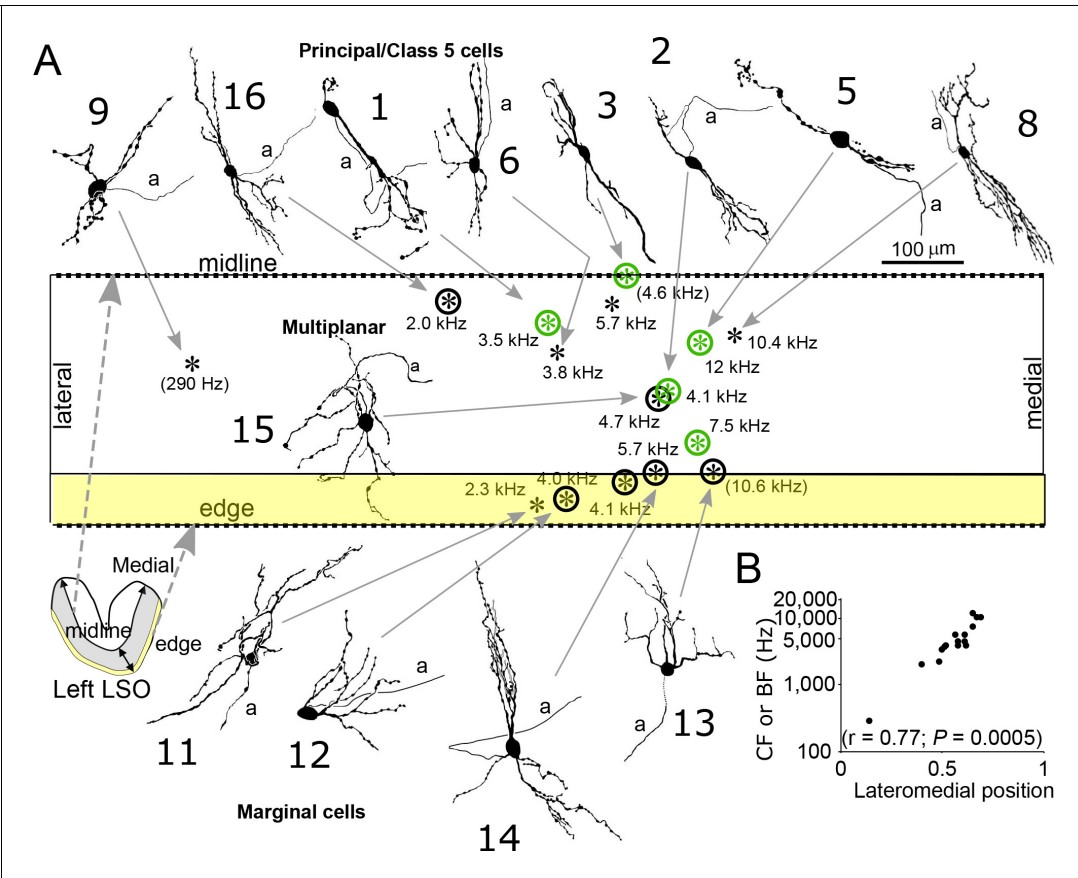

**Figure 2.** Reconstruction and location of labeled neurons. (**A**) Cell body locations (asterisks) of labeled LSO neurons are shown relative to the LSO midline and edge on a flattened lateromedial map of the LSO (compare to LSO outline in the coronal plane in the lower left corner), with indication of CF or (BF). Camera lucida drawings of neurons with a well-filled dendritic tree are shown. Numbers refer to cell numbers as in *Supplementary file 1*. Circled asterisks represent neurons for which electron microscopic analysis was done. Green symbols indicate neurons identified as principal cells at the E.M. level. The two asterisks with no arrows pointing to them represent labeled cells with lightly labeled dendritic trees that are not illustrated. (**B**) CF or BF of all 15 labeled neurons as a function of normalized lateromedial cell body location (0: lateral edge; 1: medial edge).
DOI: https://doi.org/10.7554/eLife.33854.003

consistent with earlier extracellular recordings (e.g. *Guinan et al., 1972b*). All four identified cell types conform to this tonotopic gradient.

## Principal cells are fast cells with small action potentials

Next, we asked if principal cells also differ physiologically from non-principal cells. The intracellular recordings allowed us to analyze subthreshold spontaneous activity. In all cell types, spontaneous synaptic activity consisted of excitatory (EPSPs: green dots) and inhibitory (IPSPs: red dots) postsynaptic potentials. The EPSPs were typically subthreshold. We observed that principal cells were characterized by strikingly fast, narrow EPSPs (*Figure 4A*, left column). Indeed, for principal cells EPSPs often had sub-millisecond half widths that are reminiscent of principal MSO cells (*Figure 4B*; compare with [*Scott et al., 2005*; *Franken et al., 2015*]). In contrast, EPSPs of non-principal cells were slower (*Figure 4A*, right column). Compare, for example, the train of EPSPs occurring in principal cell 1 (green dots) with the repetitive EPSPs occurring in non-principal cell 15 (green dots). In the latter, there is clear summing of the EPSPs. The number of spontaneous IPSPs (red dots) seemed to be lower than that of EPSPs, although they sometimes may have been obscured due to the high levels of EPSPs (e.g. principal cell 1, red dots show an IPSP with an intervening EPSP). In some cells IPSPs were more prominent, as seen in principal cell five and non-principal cell 16. Again, these may be visible here because the number of events was lower. We quantified the temporal properties of subthreshold postsynaptic waveforms by computing amplitude spectra and determining the upper cut-

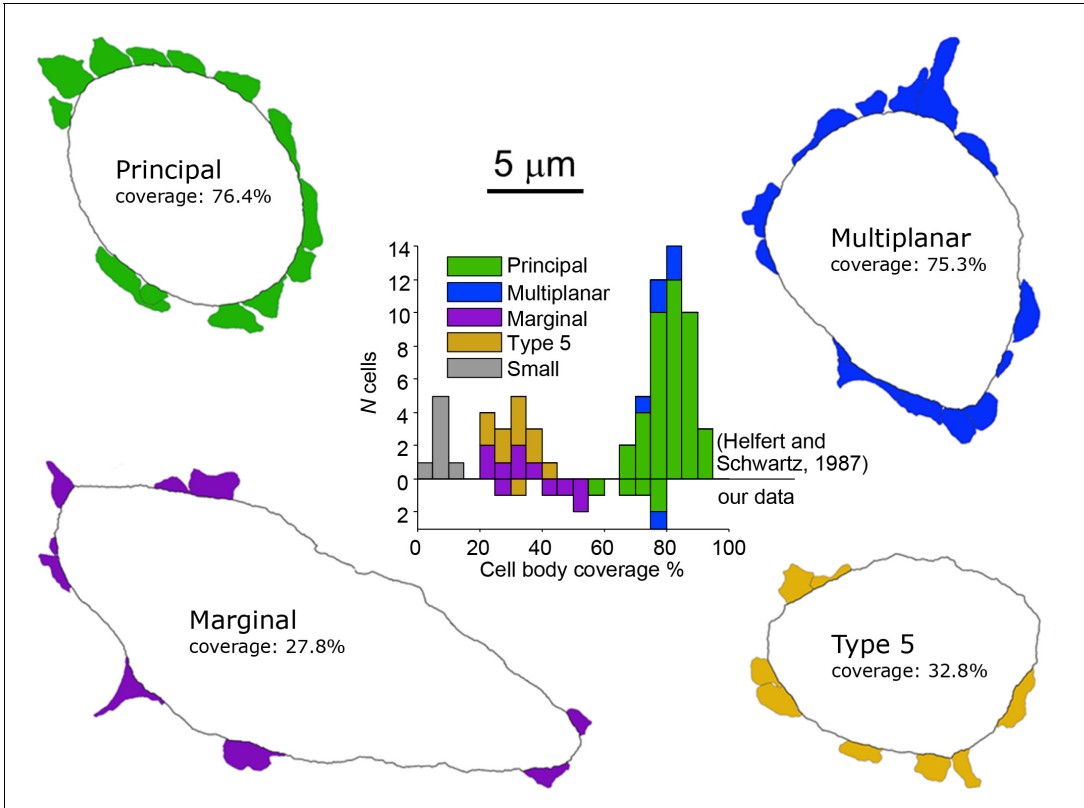

**Figure 3.** Electron microscopic analysis of cell body synaptic coverage. Electron microscopic cell body profiles of example cells of four cell types, with colored presynaptic terminals (color coded for cell type). Middle plot shows histogram of cell body synaptic coverage for LSO cell types for the data from Helfert and Schwarz (*Helfert and Schwartz, 1987*) and for our data (above and below the horizontal line at 0, respectively).
DOI: https://doi.org/10.7554/eLife.33854.004

off frequency at 50% (i.e. −6 dB) of the maximal amplitude (*Figure 4C*, intersection with horizontal dashed line). Samples (2–6 s) of spontaneous activity were high-pass filtered and wavelet de-noised (Materials and methods). Spontaneous activity of LSO principal cells contained a significantly larger proportion of high frequencies than that of LSO non-principal cells (upper cut-off frequency was respectively 528.7 ± 66.5 and 131.4 ± 15.5 Hz; two-sample t test $t_{10}$ = 6.83, p=4.6×10$^{-5}$, Hedges' g = 4.00), and was similar to spectra of MSO principal cells.

In addition to subthreshold activity, suprathreshold activity also set principal cells apart from non-principal cells. We observed that action potentials were smaller in principal cells than in non-principal cells and typically did not overshoot 0 mV (*Figure 4D*). This difference was significant across the population of labeled neurons: principal neurons had an average action potential amplitude of 28.3 ± 5.2 mV (mean ± s.e.m.), and non-principal neurons an average action potential amplitude of 66.3 ± 4.5 mV (two-tailed two-sample *t* test: $t_{10}$ = −5.48, p=2.67×10$^{-4}$, Hedges' g = 3.21). This difference could not be explained by a systematic difference in bridge balance between principal cells and non-principal cells (series resistance was 65.2 ± 6.4 MΩ for principal cells and 65.6 ± 8.1 MΩ for non-principal cells (mean ± s.e.m.), two sample $t_9$ = −0.03, p=0.98).

These two features, upper cut-off frequency of the spectrum of spontaneous membrane potential fluctuations and action potential amplitude, cluster differently for morphologically confirmed principal cells and non-principal cells (*Figure 4E*, compare green circles and black stars). Three labeled LSO neurons with morphology compatible with principal cells were not evaluated at the E.M. level, and it is therefore not possible to anatomically distinguish them from class 5 cells (*Helfert and Schwartz, 1987*). However, analysis of these physiological features shows that these cells segregate with principal cells (red crosses in *Figure 4E*; *Figure 4—figure supplement 1*; *Supplementary file 1* cells 6,7,9). Another nine intracellularly recorded neurons were not retrieved histologically, but all

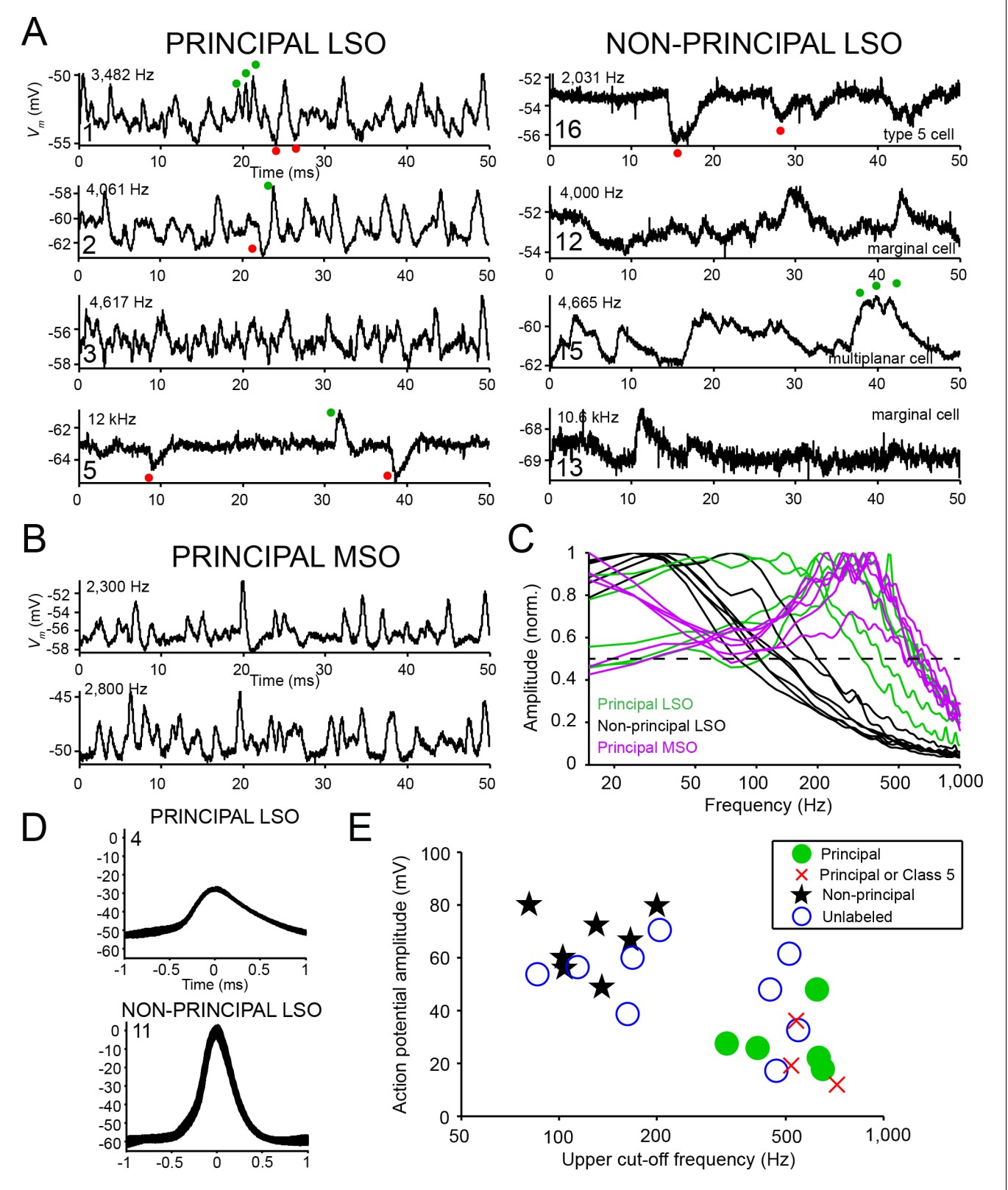

**Figure 4.** Principal cells have fast kinetics and small action potentials. (A) Fifty milliseconds of spontaneous activity is shown for four principal cells (left column) and four non-principal cells (right column). In each panel, CF or BF is indicated (top left) as well as cell number (bottom left). For the non-principal cells in the right column, the subtype is indicated above the abscissa. Red and green dots respectively indicate examples of inhibitory and

*Figure 4 continued on next page*

*Figure 4 continued*

excitatory postsynaptic potentials. (**B**) Similar as A, for two labeled MSO neurons. (**C**) Normalized amplitude spectra of spontaneous activity traces for 12 LSO neurons, as well as six labeled MSO neurons (CF ranged from 2 to 3 kHz; including the examples in B). Between 2 and 6 s of spontaneous activity were analyzed for each neuron. Dashed horizontal line indicates 50% of maximal amplitude. (**D**) Stacked action potentials during ipsilateral sound presentations for a representative principal and a non-principal cell. Cell number is indicated in each panel. Respectively 42 (top) and 407 (bottom) action potentials are overlaid. (**E**) Scatter plot of average action potential amplitude versus the upper cut-off frequency. The latter was defined as the frequency where the high-frequency flank of the normalized magnitude spectrum falls to 50% of the maximal amplitude (dashed line in panel c). Green filled circles and black stars respectively indicate morphologically unambiguous principal cells and non-principal cells. Red crosses indicate labeled neurons that light microscopically could be principal cells or non-principal class 5 cells, but were not further evaluated at the E.M. level. Blue circles indicate whole-cell recordings from unlabeled LSO neurons.

DOI: https://doi.org/10.7554/eLife.33854.005

The following figure supplement is available for figure 4:

**Figure supplement 1.** Spontaneous activity of bipolar LSO neurons.

DOI: https://doi.org/10.7554/eLife.33854.006

generated EPSPs to ipsilateral sound, IPSPs to contralateral sound and showed ILD sensitivity. Analysis of sub- and suprathreshold properties showed that 4 of these neurons cluster with the identified principal cells, and the remaining five with non-principal cells (*Figure 4E*, blue circles). In our further analyses, we consider this population of 12 principal cells and 12 non-principal cells.

## Principal cells generate onset and not chopper responses

We evaluated responses to ipsilateral tones near or at CF or BF in principal and non-principal neurons at multiple sound levels. Surprisingly, we found that ipsilateral tones generated strong onset responses in principal neurons (*Figure 5*, top), and never a sustained, chopping pattern, which is considered the canonical spiking pattern of LSO neurons (*Tsuchitani, 1977*; *Joris and Yin, 1995*; *Tollin et al., 2008*). Instead, chopping firing patterns were restricted to non-principal neurons (*Figure 5*, bottom, e.g. cells 10, 14 and the cell with CF 3.2 kHz; *Figure 5—figure supplement 1*), which showed other response patterns as well. Since none of the principal cells showed a chopper pattern or high level of sustained activity, our data imply that principal cells are severely underrepresented in the LSO literature, and that this literature builds on a biased selection of non-principal neurons. We also identified response patterns to ipsilateral tones for 7 out of 11 extracellularly recorded ILD-sensitive LSO neurons: four neurons had predominant onset spiking and three neurons generated a sustained chopping pattern (data not shown).

To relate the differences in spiking pattern to intrinsic cell properties, we obtained responses to current step injections for six neurons. We observed that all principal cells tested responded with an onset spike to depolarizing current injections, and offset spikes to hyperpolarizing responses (*Figure 6A*, top). In contrast, non-principal cells were able to generate sustained responses to depolarizing current injections (*Figure 6A*, bottom). Furthermore, the spike patterns to depolarizing current steps were strikingly similar to those elicited by ipsilateral tones for these neurons (*Figure 6B*, compare with *Figure 6A*): principal cells showed onset patterns to both sound and current injection, while sustained responses were observed in non-principal cells to both types of stimuli. We conclude that principal cells are characterized by onset firing, which relates to the intrinsic properties of the neuron, as it can be found not only to auditory stimuli but also to depolarizing current steps. Again, this is a feature reminiscent of principal MSO neurons, which also respond with a single onset spike to depolarizing current steps (e.g. *Scott et al., 2005*).

In response to hyperpolarizing current pulses we observed a voltage sag indicative of $I_h$ current in all neurons regardless of type (*Figure 6A*; note that the principal cell shown in the second row also generated such a sag, visible in a separate dataset with adjusted bridge balance (not shown)). We could calculate the input resistance at the peak and steady state of the voltage response to hyperpolarizing current in two principal cells and three non-principal cells: both principal cells had a lower input resistance than the non-principal cells (*Supplementary file 1*).

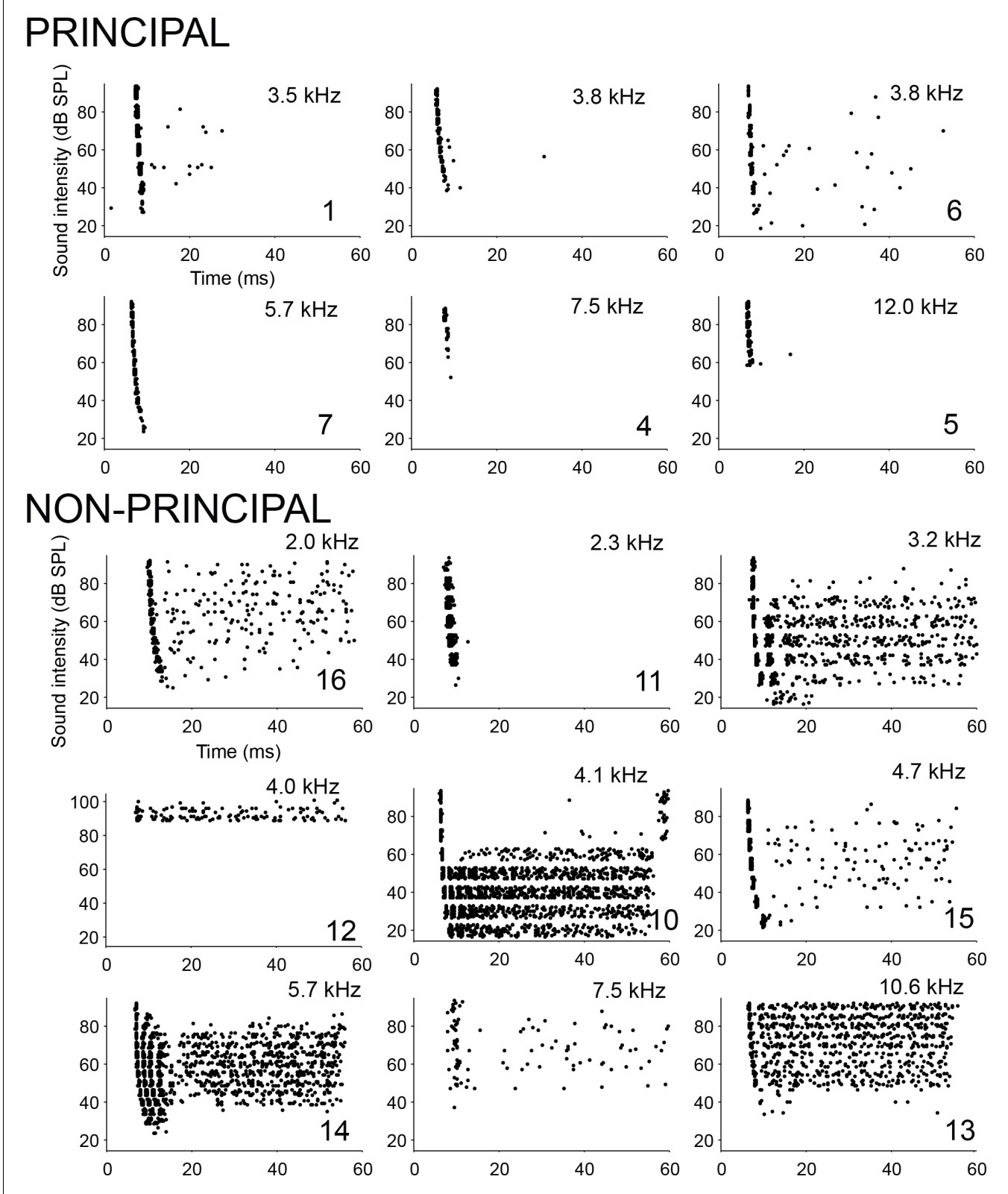

**Figure 5.** Spike patterns in response to ipsilateral tones. Dot rasters of spike responses to ipsilateral tones at or near CF or BF, for six principal cells (top) and nine non-principal cells (bottom). Each horizontal band of dots shows responses to a number of repetitions at a given SPL. CF or BF is indicated in each panel, and, for labeled cells, the cell number (*Supplementary file 1*).

DOI: https://doi.org/10.7554/eLife.33854.007

*Figure 5 continued on next page*

*Figure 5 continued*

The following figure supplement is available for figure 5:

**Figure supplement 1.** Spike patterns of labeled cells to ipsilateral tones near CF.

DOI: https://doi.org/10.7554/eLife.33854.008

## Contralaterally-evoked inhibition is graded in non-principal but not principal neurons

Our analysis of in vivo responses so far has focused on the marked differences in response to ipsilateral sound. These differences extend to effects of contralateral sound. In all labeled LSO neurons, ipsilateral tones evoked mainly EPSPs and contralateral sounds evoked mainly IPSPs. The top panels in *Figure 7A* zoom in on depolarization and the bottom panels on hyperpolarization, evoked by ipsi- and contralateral tones, respectively. As was the case for responses to ipsilateral stimulation, the response of principal cells to contralateral stimulation also showed a pronounced peak (but in the hyperpolarization direction) at sound onset, and a much smaller hyperpolarization during the sustained part of the response (*Figure 7*; *Figure 7—figure supplement 1A,B*). The latency of the inhibitory onset response decreases by several ms with increasing sound level (see next section). By comparison non-principal cells typically show postsynaptic potentials with larger amplitudes (*Figure 7*; *Figure 7—figure supplement 1A,B*), often with a more sluggish onset. These differences resulted in significantly smaller peak-to-sustained postsynaptic amplitude ratios for non-principal cells than for principal cells, during monaural ipsilateral as well as monaural contralateral stimulation (*Figure 7—figure supplement 1C*; resp. two-tailed $t$-test $t_{10} = 2.24$; p=0.049 and $t_{10} = 3.02$; p=0.013). Thus, also for the inhibitory responses, non-principal cells are more integrative than principal cells. Excitatory and inhibitory responses were matched in the sense that the evoked spike rate during ipsilateral sound stimulation correlated with the amplitude of hyperpolarization during contralateral sound stimulation (*Figure 7—figure supplement 1D*; r = −0.64; p=0.035). This could be due in part to a difference in input resistance, as suggested by the current injection data mentioned above.

The amplitude of both excitatory and inhibitory potentials typically increased with sound level (*Figure 7B*). The relation to sound level was often non-monotonic, where the amplitude decreased again for the highest sound levels (*Figure 7B*; *Figure 7—figure supplement 1A,B*). The sound level at which the non-monotonicity occurred, as well as the observation that the non-monotonicity is mirror-symmetric for ipsi- and contralateral stimulation, suggest that it may reflect acoustic crosstalk. Using a similar experimental approach in gerbil to study monaural neurons that are part of the LSO-circuit, interaural attenuation was found to usually be 50 dB or more, but occasionally it was as low as 30 dB (*Wei et al., 2017*). Indeed, in neurons with non-monotonic responses, IPSPs could be observed to high-level ipsilateral tones (*Figure 7—figure supplement 2*) and EPSPs to high level contralateral tones (*Figure 7—figure supplement 3*). Another possible mechanism for the non-monotonic response to ipsilateral stimulation is ipsilateral inhibition, which has been reported in vitro (*Wu and Kelly, 1994*) and was also inferred from in vivo observations of inhibitory sidebands (*Brownell et al., 1979*; *Caird and Klinke, 1983*). For a few neurons, we obtained responses to ipsilateral tones at various frequencies, and observed occasional hyperpolarizations suggestive of ipsilateral inhibition. In any case, the intracellular responses show that the non-monotonicity reflects neural interaction at the site of LSO neurons, and is not simply a reflection of the (more modest) non-monotonicity that can be found in the monaural inputs to LSO neurons (*Wei et al., 2017*; *Kopp-Scheinpflug et al., 2002*; *Kuenzel et al., 2011*; *Typlt et al., 2012*).

## ILD sensitivity

The different response patterns that we identified for principal and non-principal cells are reflected in functional differences in terms of ILD coding. The left panels of *Figure 8* show averaged intracellular recordings for different ILD conditions for a principal and non-principal neuron; the right panels show averaged monaural responses at sound levels matched to the ILD conditions on the left. The averaged monaural responses of the principal cell in *Figure 8A* are again a transient EPSP at the onset of an ipsilateral tone at 50 dB (right column, top panel), and transient IPSPs to contralateral tones of different sound levels. These IPSPs change little in amplitude or duration with increasing

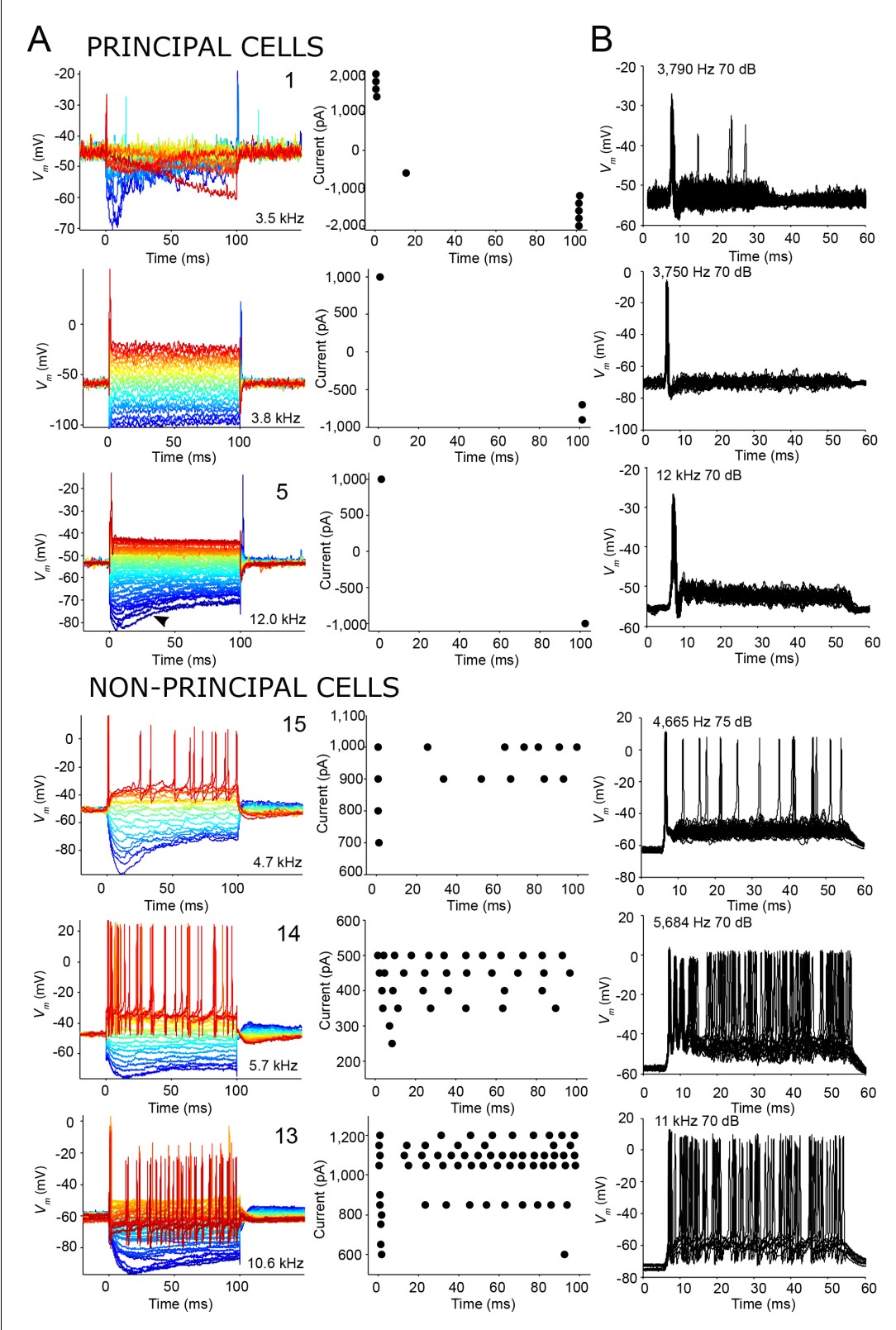

**Figure 6.** Similar spike patterns elicited by current injection and sound stimuli. (**A**) Left panels: voltage responses to depolarizing and hyperpolarizing current step injection for three principal cells (top) and three non-principal cells (bottom). 0 ms corresponds to start of current injection. Colors indicate current amplitude (blue: most negative, red: most positive), and ranged respectively from (top to bottom): −2,000 to 2,000; −1,000 to 1,000; −1,000 to 1,000; −1,000 to 1,000; −500 to 500; −1,000 to 1,200 pA. The CF or BF, and cell numbers are indicated in the panel. Arrowhead (third row, left panel)

*Figure 6 continued on next page*

eLIFE Research article

*Figure 6 continued*

indicates voltage sag indicative of I_h current that can be seen in all cell types. Right panels: spike dot rasters showing spike occurrences in the data in the left panels. (B) Stacked voltage responses to repetitions of short ipsilateral tones at or near CF/BF at one sound level, for the same cells as in A.
DOI: https://doi.org/10.7554/eLife.33854.009

SPL, but clearly decrease in latency. The binaural responses to these tones at the same respective sound levels (*Figure 8A*, left panel) suggest that the decrease in latency is an important contributor to generate ILD tuning. The latency of the EPSP evoked by the ipsilateral tone at 50 dB is similar to that of the IPSP evoked by a contralateral tone of the same SPL, and when combined (ILD = 0 dB), they interact to a response that fails to elicit a spike. However, at a contralateral sound level of 30 dB (ILD = −20 dB), the contralateral IPSP arrives simply too late to be able to suppress the ipsilateral excitation evoked by a 50 dB tone.

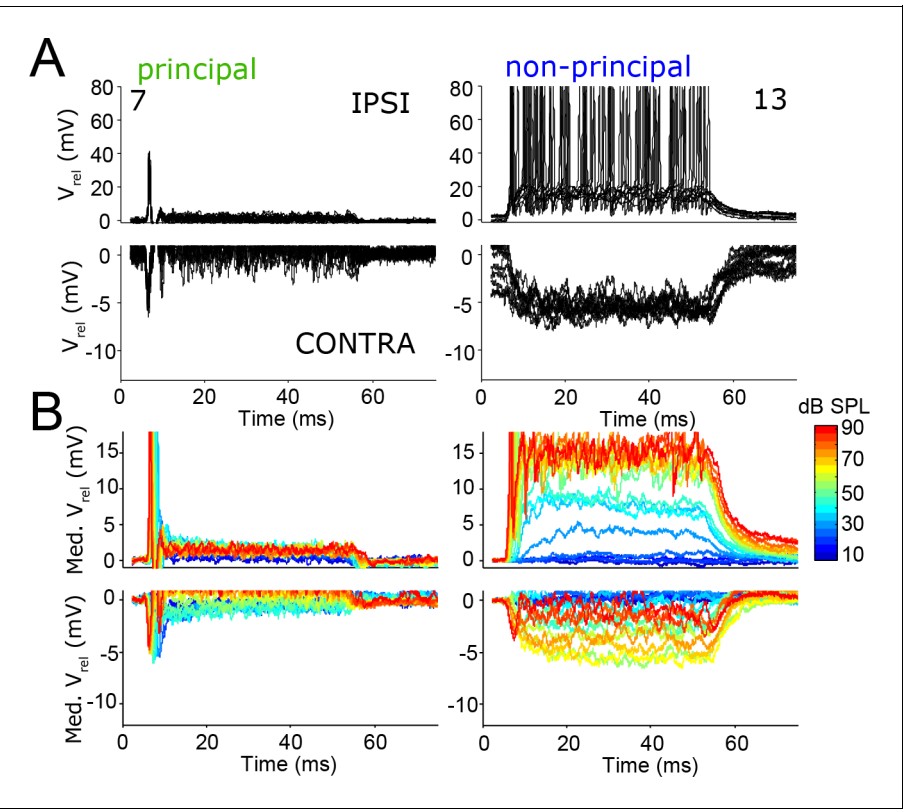

**Figure 7.** The temporal pattern of inhibition matches that of excitation. (A) Repetitions of evoked responses to monaural ipsilateral (top panels) or contralateral (bottom panels) tones at CF or BF at 70 dB SPL, for two example neurons (numbers refer to *Supplementary file 1*). (B) Median $V_m$ to ipsilateral and contralateral tones at CF or BF at different sound intensities, for the same neurons as in A. Respective stimulus frequencies and number of repetitions: 5.6 kHz, N = 20 (left column); 11 kHz, N = 10 (right column).
DOI: https://doi.org/10.7554/eLife.33854.010

The following figure supplements are available for figure 7:

**Figure supplement 1.** Sound-evoked excitation and inhibition is balanced.
DOI: https://doi.org/10.7554/eLife.33854.011

**Figure supplement 2.** Stacked responses of a non-principal LSO cell to ipsilateral tones at CF (5.7 kHz).
DOI: https://doi.org/10.7554/eLife.33854.012

**Figure supplement 3.** Stacked responses of the same cell as in *Figure 7—figure supplement 2* to contralateral tones at CF (5.7 kHz).
DOI: https://doi.org/10.7554/eLife.33854.013

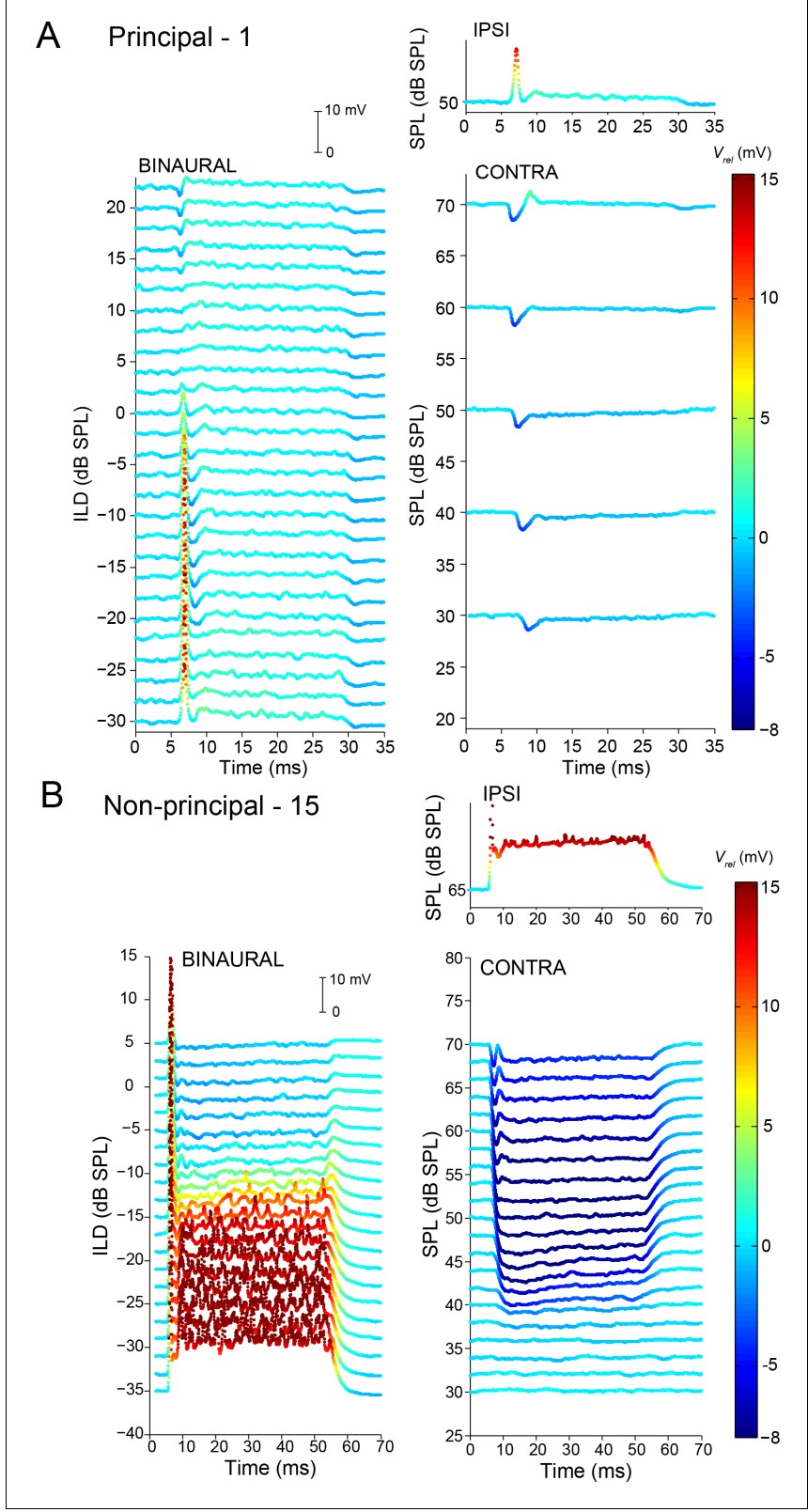

**Figure 8.** Complementary ILD-sensitivity in principal and non-principal neurons. Average responses to binaural and monaural tones are shown for a principal cell (**A**) and non-principal cell (**B**). Average $V_m$ is color-coded relative to resting $V_m$. Left panels: average responses to binaural tones of varying ILD, where the contralateral sound level is varied and the ipsilateral sound level kept constant. Right panels: average responses to monaural tones at the

*Figure 8 continued on next page*

*Figure 8 continued*

same sound levels as for the binaural data in the left panels. Respective stimulus frequencies: A: 3,480 Hz; B: 4,665 Hz.

DOI: https://doi.org/10.7554/eLife.33854.014

The following figure supplement is available for figure 8:

**Figure supplement 1.** Complementary ILD-sensitivity in principal and non-principal neurons.

DOI: https://doi.org/10.7554/eLife.33854.015

Because non-principal neurons have slower PSPs (*Figure 4*) and sustained responses to ipsi- and contralateral sound, their spike output is less dependent on the exact timing of PSPs, and ILD-sensitivity can extend throughout the stimulus duration. The non-principal cell illustrated in *Figure 8B* displays ILD-sensitivity which is opposite to that of the principal cell in *Figure 8A*. Here, sustained inhibition builds up in time to a level where it effectively overcomes sustained excitation throughout the duration of the stimulus, even at slightly negative ILDs where the sound level to the excitatory ear is higher. However, the inhibition is not able to suppress the onset response.

As is the case for their monaural responses (*Figure 5*), the responses of non-principal cells to ILDs was quite varied. The pattern shown in *Figure 8B* is consistent with spike patterns observed in extracellular studies where the onset response is not or not completely inhibited (*Sanes and Rubel, 1988*). But in many cells, both in published extracellular recordings of cells with sustained responses and in our identified non-principal cells, ILD-sensitivity covers both onset and sustained portions of the response (*Figure 8—figure supplement 1B*).

A population analysis of ILD functions reveals indeed that both principal and non-principal cells contribute to ILD-sensitivity at sound onset (*Figure 9A*), but that principal cells do not contribute to ILD sensitivity beyond the first 10 ms (*Figure 9B*). We did not observe significant differences between principal cells and non-principal cells in terms of position of the slope of the ILD function

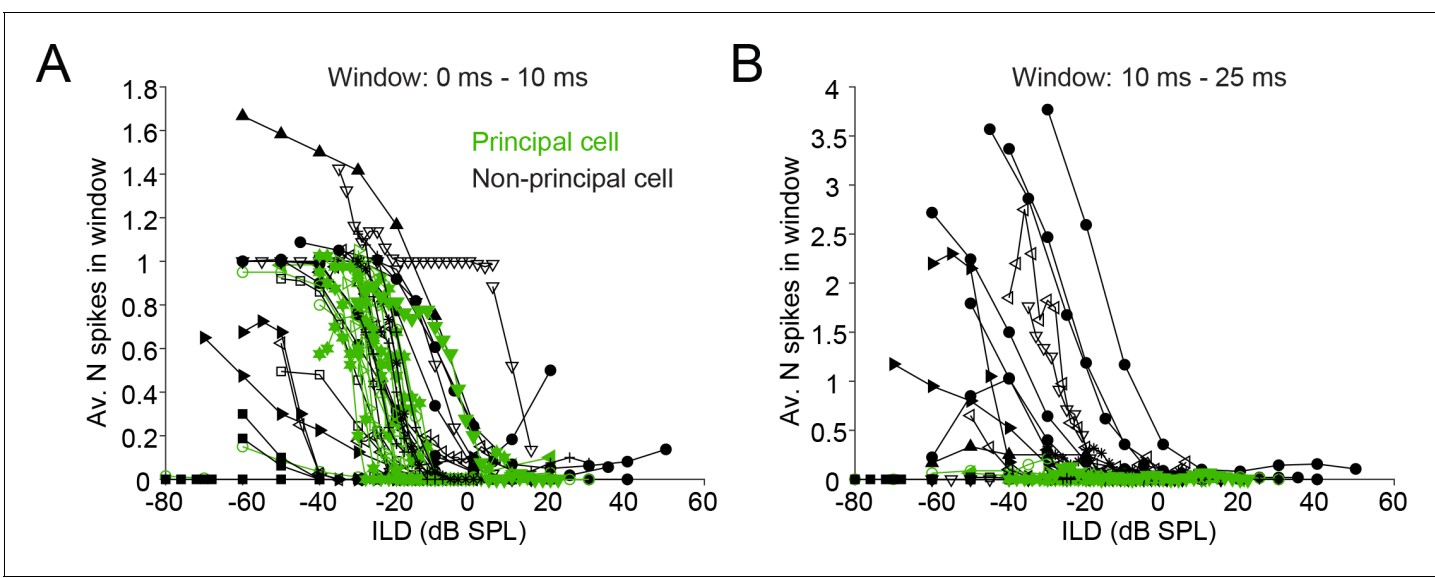

**Figure 9.** Characterization of ILD-functions in principal and non-principal neurons. (**A**) ILD functions for sound onset (from 0 to 10 ms post-stimulus onset). By convention, negative (positive) values for ILD indicate a higher sound intensity at the ipsilateral (contralateral) ear. (**B**) ILD functions for the sustained part of the stimulus. The window is limited from 10 to 25 ms post-stimulus onset because the shortest stimulus duration used was 25 ms.

DOI: https://doi.org/10.7554/eLife.33854.016

The following figure supplements are available for figure 9:

**Figure supplement 1.** Metrics of ILD functions.

DOI: https://doi.org/10.7554/eLife.33854.017

**Figure supplement 2.** Level-dependence of ILD functions.

DOI: https://doi.org/10.7554/eLife.33854.018

(*Figure 9—figure supplement 1A*) or of the range of ILDs covered by the slope (*Figure 9—figure supplement 1B*). We obtained several cells with ILD functions positioned towards quite negative ILDs compared to the gerbil's physiological range (*Maki and Furukawa, 2005*) and published gerbil LSO recordings (*Sanes and Rubel, 1988*): for 6 out of 14 cells, the midpoint of the ILD functions' slope was more negative than −20 dB ILD for all data sets obtained (*Figure 9—figure supplement 1*), compared to ~6 out of 47 cells in Sanes and Rubel (their Figure 15). This difference may be due to higher ipsilateral sound levels in our data set compared to their data: we find that higher ipsilateral sound levels can move the slope of the ILD function towards more negative ILDs, as has been previously reported (*Figure 9—figure supplement 2*; *Park et al., 2004*; *Tsuchitani and Boudreau, 1969*; *Tsai et al., 2010*). Another possibility is that there was some conductive hearing loss associated with the recording, which necessarily exposed the middle ear space (see Materials and methods). We were careful to symmetrize the exposure of the bullas and to keep the middle ears clear, but we cannot exclude asymmetries (e.g. seepage of CSF into the ipsilateral hidden dorsal middle ear cavity) causing an attenuation of the ipsilateral sound level, which would cause a shift of ILD functions towards negative ILDs. Note however that systematic threshold differences were not observed in a study of monaural fibers in the trapezoid body which used a similar recording approach (*Wei et al., 2017*).

Taken together, the monaural and binaural responses to tones (*Figures 4–9*) suggest an important functional difference for principal and non-principal neurons. Non-principal neurons integrate over time and enable the coding of ILDs of sustained sounds. Principal neurons are differentiators that are particularly sensitive to timing aspects of the sound stimulus such as stimulus onset, but lack sensitivity to sustained parts of the stimulus.

## Principal and non-principal LSO neurons both contribute projections to the ipsilateral midbrain

LSO projections are critical to characterize as they contribute to the 'acoustic chiasm' which creates the representation of contralateral space at the level of midbrain and higher (*Glendenning and Masterton, 1983*). LSO neurons are known to project both ipsi- and contralaterally, and the laterality of projection is biased in frequency and sign (excitatory vs. inhibitory). However, the relationship of laterality to other physiological properties is unknown. For 8 cells, we could follow the axon to the point where we were confident about whether they projected to the ipsilateral or contralateral inferior colliculus (*Supplementary file 1*). The three traceable principal cell axons all headed ipsilaterally as did three non-principal cells. One other non-principal cell headed contralaterally and another one projected bilaterally, bifurcating just after exiting the LSO dorsally. In addition, two labeled principal and one non-principal neuron, all of which projected to the ipsilateral inferior colliculus and are thus probably glycinergic, sent an axon collateral to the ipsilateral lateral nucleus of the trapezoid body (LNTB; one of these cells has been reported before [*Franken et al., 2016*]). Three non-principal neurons had local axon collaterals in the LSO (*Figure 10* and Supplementary Text). In summary, the data do not support a simple dichotomy between principal and non-principal cells in laterality of projections to the midbrain or superior olivary complex.

## Discussion

We obtained in vivo intracellular recordings from morphologically identified neurons of all four large cell categories in LSO. Our findings confirm the morphological classes discerned in this species (*Helfert and Schwartz, 1987*) and allow us to link these to physiology. We present the first evidence that all these cell types are excited by sounds to the ipsilateral ear and inhibited by sounds to the contralateral ear, and that they conform to the tonotopic LSO map. Unexpectedly and most importantly, our data reveal that principal LSO cells do not show the prototypical integrative chopper responses that have been attributed to these cells since the earliest single-cell studies: rather, they display fast kinetics and onset responses, which necessitates a revision of the prevailing notion of LSO as a high-frequency nucleus integrating auditory information over time to send ILD cues to higher centers in the brain. Axons of several principal and non-principal cells could be followed into the ipsilateral or contralateral lateral lemniscus (LL) indicating that these different cell types are olivo-collicular. In addition, local axon collaterals were occasionally seen, as well as collaterals heading into the LNTB, and we suggest that the latter projection is glycinergic (Supplementary Text). LSO

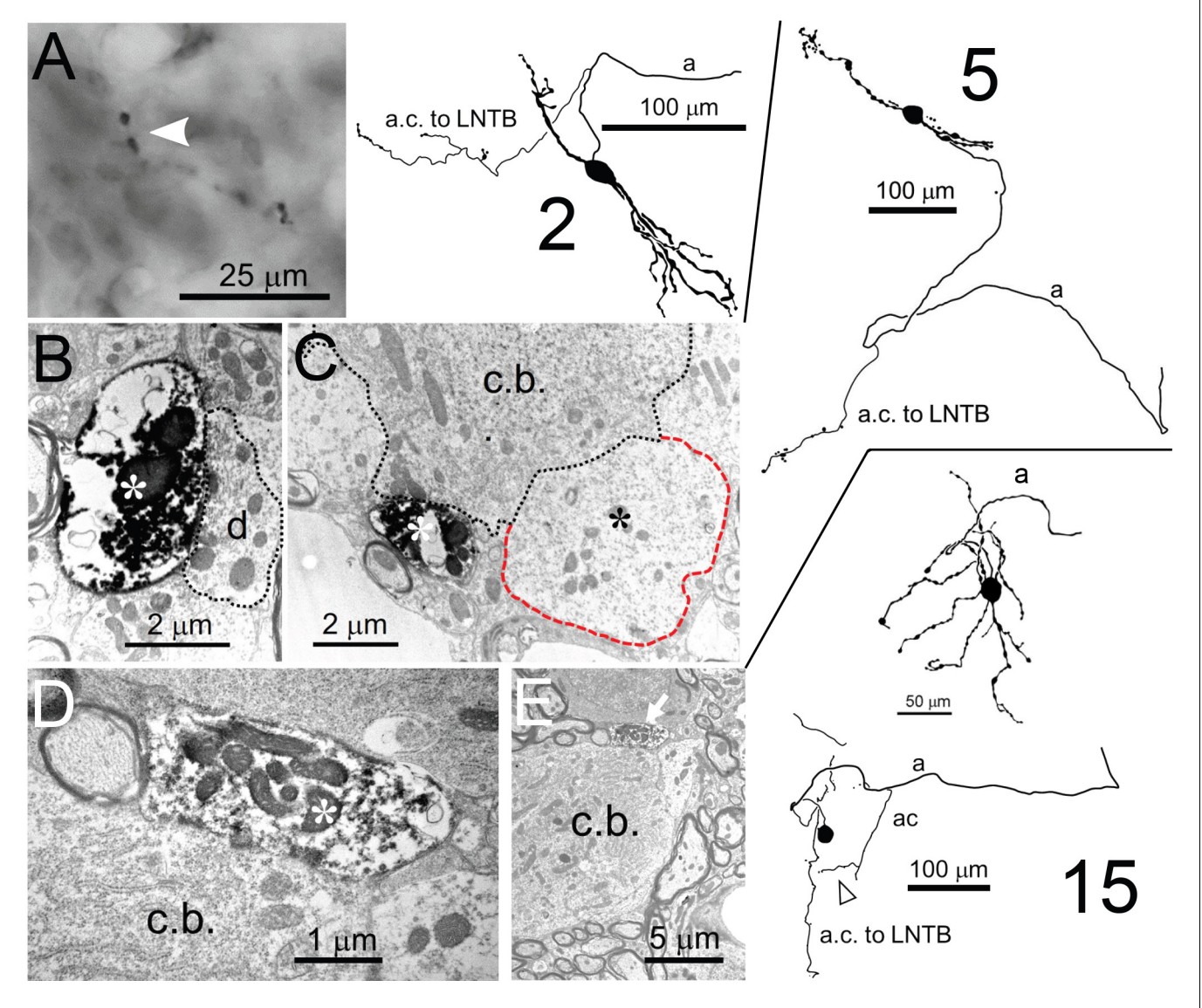

**Figure 10.** Collateral projections of LSO neurons. Camera lucida drawings of 3 neurons are shown on the right: the cell number refers to the *Supplementary file 1*. Corresponding E.M. micrographs are shown on the left. (**A**) Labeled axon collateral in the LNTB from a principal LSO cell. It has large swellings suggestive of presynaptic terminals. a: axon; a.c.: axon collateral. (**B,C**) E.M. images of a labeled presynaptic terminal (white asterisk) from another LSO principal cell, making synaptic contact with dendritic (**B**) and somatic (**C**) compartments of an LNTB neuron. Red outline/black asterisk indicates an unlabeled large presynaptic terminal with characteristics typical of a globular bushy cell terminal. d.: dendrite; c.b.: cell body. (**D,E**) E.M. images of a labeled presynaptic terminal (white asterisk in D, white arrow in E) of a local axon collateral in the LSO on a cell body (c.b.). The collateral is from a non-principal multiplanar cell. Top camera lucida drawing for neuron 15 is the cell body with dendritic tree and part of the axon, bottom camera lucida drawing is the same cell body, without dendrites but with axon and axon collaterals. Arrowhead below bottom camera lucida drawing indicates the collateral whose swelling is shown in the E.M. images.
DOI: https://doi.org/10.7554/eLife.33854.019

principal cells provide a computational nexus in parallel to MSO principal cells. Traditionally, these two circuits are thought to mainly provide high-frequency ILD and low-frequency ITD information, respectively, to higher centers. MSO cells sense microsecond differences in arrival time using $I_{LVK}$ and $I_h$ currents that lead to low input resistance, fast postsynaptic potentials and onset spikes to depolarizing current injection (*Scott et al., 2005*; *Mathews et al., 2010*; *Khurana et al., 2011*; *Franken et al., 2015*). LSO neurons are biased to high frequencies, where ongoing temporal cues are less dominant. We show that LSO principal cells have several properties reminiscent of MSO

principal cells. They display smaller spikes than non-principal cells, and consistently fire onset spikes to positive current injection. Firing patterns evoked by sustained auditory stimuli mirror responses evoked by current injection: principal cells generate an onset spike to sustained tones, unlike non-principal cells which typically fire throughout the stimulus duration. This difference in spiking activity was accompanied by contrasting postsynaptic potential profiles with mainly onset responses in principal cells and more sustained activity in non-principal cells. These are unexpected properties that are inconsistent with the prevailing view that LSO principal cells integrate auditory stimulation over time by generating sustained, chopping responses to tones.

The traditional association of ILD-sensitive neurons in LSO with chopper responses, which has stood for 50 years, emerged from pioneering studies in the cat reporting the first single-cell recordings (*Tsuchitani and Boudreau, 1966*, *1968*, *Boudreau and Tsuchitani, 1970*; *Guinan et al., 1972a*, *1972b*; *Tsuchitani, 1977*). Not only did these early studies find chopper responses to be the most frequently observed; phasic and transient responses were considered outright pathological (*Tsuchitani and Johnson, 1991*; *Boudreau and Tsuchitani, 1970*). This point of view biased the ensuing literature. For example, in an early review these authors commented: "We have frequently found that some electrodes will only record phasic units. The smaller the active area of the electrode tip the greater the likelihood that phasic units will be encountered' (*Boudreau and Tsuchitani, 1970*), and they undertook technical steps to decrease the proportion of phasic cells recorded from in their experiments. In later studies, the presence of ILD-sensitive choppers in LSO was confirmed, not only in cat (*Caird and Klinke, 1983*; *Joris and Yin, 1995*; *Tollin et al., 2002*; *Greene and Davis, 2012*) but also in other species, for example in chinchilla (*Finlayson and Caspary, 1989*), rat (*Irvine et al., 2001*), bat (*Harnischfeger et al., 1985*), and gerbil (*Sanes and Rubel, 1988*). The chopper response came to be seen as a signature property, even though a minority of other response types was also reported, including transient responses. Because the principal cells are the most numerous cells in LSO, it was reasonable to assume that these cells produce the most frequently encountered response type, that is chopping. In none of these studies, however, was single-cell labeling attempted, except in one study in chinchilla (*Finlayson and Caspary, 1989*). Again, in this study the initial assumption was that '... LSO principal cells ... displayed chopper responses to ipsilateral stimuli'. Two cells with chopping responses were labeled with Lucifer yellow-filled electrodes and were described as 'principal' because of the bipolar nature of their dendrites (e.g. their *Figure 4*). This cell was 90–100 μm from the edge of the LSO making it likely that it is instead a marginal cell with a dendritic tree similar to two of our marginal cells illustrated in our *Figure 2* (D. Caspary, personal communication, October 2017).

In the MSO, small somatic spikes are (along with a very large field response) a contributing factor to the well-known difficulty in recording from these cells with extracellular techniques. Our finding that LSO principal cells have fast membranes and small action potentials (*Figure 4*) suggests that these cells are also less easily isolated with extracellular techniques, compared to non-principal cells with larger overshooting spikes. This may explain the relative paucity of onset responses and preponderance of other response types reported in LSO. Also, the small action potentials of principal cells may necessitate close physical proximity for single-cell isolation with metal electrodes, and perhaps explains the previously reported association with injury potentials and pathology leading to underrepresented reporting of these responses in the literature.

Our finding has far-reaching functional implications and provides a simplifying view not only on discrepancies in the literature but also on well-documented but previously puzzling features of the LSO-circuit. 'Chopping' refers to regular firing where the time of occurrence of output spikes is not determined by the time of occurrence of input spikes. This behavior has been particularly well-studied in the cochlear nucleus, where it is associated with cells that have slow membrane time constants and show poor coding of the temporal fine-structure of sounds (*Young and Oertel, 2004*). Because sound intensity is not an instantaneous property and its measurement inherently requires some temporal integration, the finding of neurons with chopper responses in a nucleus computing ILD makes computational sense. However, the striking morphological and physiological specializations that are found in this circuit, most dramatically in its inhibitory limb in the form of the calyx of Held in the MNTB (*Figure 1A*), do not fit the role of ILD computation but suggest a need for temporal precision (*Joris and Yin, 1995*). Spherical bushy and MNTB principal cells primarily receive very large individual suprathreshold somatic synapses and at high frequencies they respond very similar to sound as their inputs (*McLaughlin et al., 2008*). Globular bushy cells receive many smaller subthreshold

somatic and/or dendritic inputs and, as expected from their 'fast' membranes, respond reliably when the subthreshold inputs arrive at the same time, as for example at sound onset. The importance of timing in the LSO is also supported by modeling work, showing that precise temporal processing is necessary to reproduce realistic responses to amplitude-modulated sounds (*Ashida et al., 2016*). Our finding that principal cells are not the temporal integrators they were thought to be, but are actually more akin to the MSO and the afferents to LSO and are triggered by fast changes in membrane potential, suggests that this nucleus has a dual role: it is involved in both ILD and temporal processing.

The exact nature of this temporal role requires further work, but one promising observation is the steep sensitivity to ITDs of sound transients found in LSO (*Caird and Klinke, 1983*; *Joris and Yin, 1995*; *Irvine et al., 2001*). Our data also suggests that the temporal precision of LSO neurons can contribute to their ILD tuning, because the latency of postsynaptic potentials decreases when stimulus intensity increases (*Figure 8A*). Such time-intensity trading has been described in in vitro (*Sanes, 1990*) and in vivo studies from the LSO and other binaural nuclei, especially for transient stimuli (*Pollak, 1988*; *Irvine et al., 2001*).

Our finding of fast membrane features in LSO principal cells suggests that discrepancies in the in vitro literature are based on the lack of cell identification. Two studies using mouse and/or gerbil slices (*Hassfurth et al., 2009*; *Walcher et al., 2011*) indicated that, similar to our observations, virtually all LSO principal cells in both species responded to depolarizing current steps with an onset response. They suggested that other non-onset responses from LSO cells described in other reports (see below) were due to the young age of the animals used, varying ACSF/electrode solutions and unnatural bath temperatures. It is, however, also possible that some of the recordings in these reports were from non-principal cells because in most cases the cells were not identified anatomically (*Sanes, 1990*; *Wu and Kelly, 1994*; *Kandler and Friauf, 1995*; *Sterenborg et al., 2010*; *Kotak and Sanes, 2014*; *Remme et al., 2014*) and many of the cells described had overshooting action potentials, a feature that in our data is confined to non-principal cells. Another factor that has been brought forward are spatial gradients, along the tonotopic axis of LSO (*Koch and Sanes, 1998*). For example, Remme et al. describe in vitro data showing faster membrane properties for cells in the lateral, low frequency, limb of the LSO than for cells in the medial, high frequency, limb of the LSO (*Remme et al., 2014*). Many of the slower cells were located near the nucleus edge (their *Figure 2B*), supporting the hypothesis that these are marginal cells. Labeled LSO neurons combined with firing properties to depolarizing current pulses have been reported in three papers (*Kandler and Friauf, 1995*; *Adam et al., 1999*; *Barnes-Davies et al., 2004*). All of these reports used rats and all of the recordings associated with labeled cells were from 16 day or younger rats. In all three cases some cells fired repetitively and others at onset only. In only one case (*Barnes-Davies et al., 2004*) were responses from cells in older slices reported (n = 6) and three were onset, three fired repetitively but no anatomy was provided for any of these. Reports indicate that the onset of hearing in rats begins are around post-natal day 14 (P14) where early peaks in auditory brainstem evoked responses (ABRs) begin to appear but do not fully mature until P22 (*Geal-Dor et al., 1993*). Thus, these recordings were probably made from a system still undergoing considerable developmental change. Our in vivo finding of onset responses in confirmed principal cells necessitates a reappraisal of the in vitro studies. We hypothesize that the presence of multiple cell types combined with various (unavoidable) sampling biases has led to discrepancies in that literature, and that it can be resolved by identification of the cells recorded from.

Most of our labeled LSO neurons were localized in the most ventral part of the nucleus, which is a consequence of the experimental approach: the patch electrode entered the brainstem ventrolaterally, so the first neurons encountered would be the most ventral ones in the nucleus. After recording from a neuron, typically no further penetrations were made to avoid damaging the labeled neuron. This explains the paucity of very low and high-CF cells in our sample. This issue probably also led to the relatively large proportion of marginal cells and small proportion of principal cells in our sample compared to that of Helfert and Schwartz (*Figure 3*).

At a broader level, our results contribute to an emerging picture of the auditory brainstem composed of parallel circuits operating at different time scales. The presence of multiple types of projection neurons extracting different cues from the auditory nerve, including different temporal features (fine-structure, envelopes, transients), has been a longstanding theme in the cochlear nucleus (*Young and Oertel, 2004*). Recent work, including the present results, suggests that even within the

different circuits originating in the cochlear nucleus and ultimately feeding into the midbrain, subcircuits are embedded that repeat the theme of using the same afferent inputs towards different computations. Interestingly, psychophysics-based models also suggest several operational time scales that differ strikingly between perceptual tasks in both monaural and binaural hearing (*Viemeister and Plack, 1993*). Further probing of the binaural sensitivity of different cell groups within and across nuclei with regard to time scale of integration, and relating this to behavioral sensitivity promises a better understanding of their mutual relationship (*Brown and Tollin, 2016*).

## Materials and methods

The methods for surgery, in vivo patch clamp recording and light and electron microscopic histology have been previously described (*Franken et al., 2015*; *2016*) and are briefly reviewed here.

### Animals and surgery

We used adult and juvenile (P28–P36) Mongolian gerbils (*Meriones unguiculates*) of either sex. The animals had no previous experimental history and were housed with six or fewer per cage, with a 10 hr light/dark cycle (lights turn on at 7 a.m., and off at 9 p.m.). This study was performed in strict accordance with the recommendations in the Guide for the Care and Use of Laboratory Animals of the National Institutes of Health. All procedures were approved by the KU Leuven Ethics Committee for Animal Experiments (protocol numbers P155/2008, P123/2010, P167/2012, P123/2013, P005/2014). The animals were anesthetized by intraperitoneal injection of ketamine (80–120 mg/kg) and xylazine (8–10 mg/kg) in NaCl 0.9%. Maintenance of anesthesia was ensured by additional intramuscular injections of ketamine (30–60 mg/kg) and diazepam (0.8–1.5 mg/kg) *in aqua*, guided by the toe pinch reflex. Body temperature was monitored and maintained at 37°C with a homeothermic blanket (Harvard Apparatus, Holliston, MA, USA) and a heating lamp positioned above the animal. A transbulla craniotomy was made to expose the ventrolateral brainstem. The pinna folds were removed around the external acoustic meatus. The bulla on the contralateral side was opened as well to maintain acoustic symmetry. Meningeal layers overlying the exposed part of the brainstem were removed before electrode penetration. CSF leakage caused by opening of the meninges was wicked up or aspirated.

### Electrophysiology

Patch clamp pipettes filled with internal solution were used to obtain blind in vivo patch clamp recordings. Pipettes were pulled from borosilicate capillaries (1B120F-4, World Precision Instruments, Inc., Sarasota, FL, USA) with a horizontal puller (Model P-87, Sutter Instrument Co., Novato, CA, USA). Electrode resistances were 5–7 MΩ when filled with internal solution and measured in cerebrospinal fluid. The internal solution contained (in mM) 115 K gluconate (Sigma); 4.42 KCl (Fisher); 10 $Na_2$ phosphocreatine (Sigma); 10 HEPES (Sigma); 0.5 EGTA (Sigma); 4 Mg-ATP (Sigma); 0.3 Na-GTP (Sigma); and 0.1–0.2% biocytin (Invitrogen). pH was brought to 7.30 with KOH (Sigma) and osmolality to 300 mmol/kg with sucrose (Sigma). A patch clamp amplifier (BC-700A; Dagan, Minneapolis, MN, USA) was used to obtain membrane potential recordings, where the analog signal was low-pass filtered (cut-off frequency 5 kHz) and digitized at 50–100 kHz (ITC-18, HEKA, Ludwigshafen/Rhein, Germany; RX8, Tucker-Davis Technologies, Alachua, FL, USA). Series resistance was $61.3 \pm 3.3$ MΩ (mean ± SEM; $N = 23$, excluding one outlier with a series resistance >100 MΩ). Opening resting membrane potential was $-56.3 \pm 0.69$ mV (mean ± SEM, $N = 23$).

### Stimuli

Physiological recordings were done in a double-walled sound-proof booth (IAC, Niederkrüchten, Germany). Tucker-Davis Technologies System II hardware was controlled by MATLAB scripts to generate and present sound stimuli. Etymotic speakers attached to hollow ear bars were positioned over the ears. The system was acoustically calibrated before each experiment with a probe microphone (Bruel and Kjaer, Nærum, Denmark). Frequency tuning was measured using a threshold-tracking algorithm during ipsilateral short tone presentation, where either spikes or large EPSPs were used as triggers. Next, responses to repetitions of short tones were typically collected at the characteristic frequency, presented monaurally ipsilaterally and contralaterally (typical parameters: 50–100 ms stimulus duration, 150–200 ms interstimulus interval, 30–50 repetitions at each SPL which was

varied in steps of 5 or 10 dB). ILD sensitivity was then evaluated using short tones at CF where the ipsilateral sound level was kept constant and the contralateral sound level varied. In between blocks of stimulus-evoked responses, spontaneous activity was collected. For some cells, responses were also recorded to hyperpolarizing and depolarizing current step injections (duration 100 ms; current amplitude was varied in steps of 100–200 pA).

## Analysis

Electrophysiological data were acquired using MATLAB (The Mathworks, Natick, MA, USA) and Igor-Pro (WaveMetrics, Lake Oswego, OR, USA), and analyzed with custom-made scripts in MATLAB (The Mathworks, Natick, MA, USA). Software used for non-trivial analyses is shared in a source code file accompanying this paper (Source code 1). Membrane potential values were corrected for the liquid junction potential by subtracting 10 mV from the measured potential (*Roberts et al., 2014*). Steady-state and peak input resistances were derived from voltage responses to hyperpolarizing current steps by calculating respectively the median membrane potential during the last 10% of the step and the minimal membrane potential during the step response. The frequency spectrum of spontaneous activity (*Figure 4C/E*) was calculated by discrete Fourier transform after subtracting the median $V_m$ of the first 10 ms from the data, high-pass filtering (FIR-filter implemented in MATLAB with cut-off frequency 10 Hz) and wavelet denoising the data (MATLAB function *wden*, with parameters 'heursure', soft thresholding, no rescaling, with a 'sym8' wavelet at level 5). Action potential amplitude (*Figure 4E*) was defined as the difference between $V_m$ at action potential peak and the median $V_m$ during the first 2 ms of the same recording file (before onset of the first stimulus repetition). A regularity analysis was performed on the spike responses to monaural tones at CF (*Figure 5—figure supplement 1*), using the exact method of Wright et al. (*Wright et al., 2011*). Interspike interval statistics were only calculated for bins for which at least 20% of repetitions contained an interspike interval. To evaluate subthreshold sound-evoked response at sound onset (5–15 ms post stimulus onset) and during the sustained part of the response (40–50 or 20–25 ms), only repetitions without spikes in these intervals were included (*Figure 7—figure supplement 1*).

## Histology and electron microscopy

At the end of the experiment, the animal was overdosed with pentobarbital and perfused through the heart with saline followed by paraformaldehyde (PFA) 4% in 0.1M $PO_4$ buffer or PFA 1%/glutaraldehyde 1% and PFA 2%/glutaraldehyde 1%. Tissue processing methods for light and electron microscopy have been described previously (*Smith et al., 2005*; *2010*) and are briefly summarized here. The brain was removed and stored refrigerated in PFA 1%/glutaraldehyde 1% for at least 24 hr. 70 µm thick sections of the brainstem were then cut with a vibratome and the biocytin tracer visualized using the DAB-nickel/cobalt intensification method (*Adams, 1981*). Sections were rinsed in phosphate buffer and these free-floating sections were inspected with a light microscope to determine the location of the labeled cell, its axon and dendritic tree.

Some of the sections containing the labeled cell body and relevant portions of its dendritic tree and axon were selected to be processed for electron microscopy (E.M.). Sections not selected for E. M. were mounted on slides, dehydrated, Nissl-stained with cresyl violet and coverslipped for light microscopic evaluation. The outlines of the SOC nuclei in light microscopic images were determined using the Nissl stain. Those sections selected for E.M. analysis were fixed in 0.5% osmium tetroxide for 30 min, rinsed, and dehydrated through a series of graded alcohols and propylene oxide. Sections were then placed in unaccelerated Epon-Araldite resin and then transferred into a fresh batch of unaccelerated resin overnight. The sections were then embedded and flat mounted in accelerated resin between Aklar sheets at 65°C. The region of the plastic-embedded sections containing the labeled portion of the neuron was cut out of the 70 µm section and mounted on the flattened face of a plastic beam capsule. The 70 µm section was re-sectioned into 3 µm sections that were placed on a glass coverslip. The section containing the labeled portion of the cell was selected and removed from the glass coverslip and remounted on a beam capsule. A camera lucida drawing of the section face including the location of the labeled cell part was made and 70 to 80 nm thin sections were then cut and mounted on coated nickel grids. These thin sections were then stained with uranyl acetate and lead citrate and examined using a Philips CM-120 electron microscope.

Measurements of somatic and axon terminal features from electron micrographs were made using ImageJ software (NIH). One feature that distinguishes cell type in the gerbil LSO is the amount of somatic synaptic coverage (*Helfert and Schwartz, 1987*). To determine percentage of synaptic coverage of a cell body, the length of the surface of the labeled cell was first measured and then the length of apposition of synaptic terminals on the labeled structure was measured in at least three sections and averaged. The circumference of each synaptic terminal was also measured.

In the gerbil the main body of the LSO in coronal sections is more U-shaped than S-shaped (as seen in the cat) and has been described as resembling a duck, with distinct lateral and medial limbs (the tail and body, the U) and a very small medial limb (the head). In cresyl-violet coronal sections of the gerbil brainstem, the LSO can be distinguished as a lateral concentration of cells in the superior olive (SOC) with quite distinct boundaries due to the surrounding paucity of cells. In osmium-fixed plastic embedded sections the gerbil LSO is seen as a distinctly lighter, laterally situated SOC region surrounded by a darker neuropil, due to the higher concentration of myelinated axons there, again creating quite distinct boundaries. Using these features as boundaries a camera lucida drawing of the labeled cell body's location within the boundaries of the LSO seen in that section was made. To determine the location of each labeled LSO cell (see *Figure 2A*) within the LSO boundaries and to make comparisons across animals the following steps were taken. A line was drawn, following the contour of the LSO, from the lateral-most extend to the medial-most extent down the LSO center-line (see inset in *Figure 2A*). The length of the typically U-shaped line was measured and the location of the labeled cell body was determined relative to its position along the line. As seen in *Figure 2*, all the labeled cells were place on a generic straightened LSO with the lateral most LSO point to the left and the medial most point on the right.

Another criterion for cell type classification in the LSO is the soma's location relative to the LSO edge/border. As can be seen in *Figure 2*, cells varied in location relative to the curved dorsal and ventral edges, so for each cell a perpendicular line was drawn from the midline through the cell to the LSO edge and the relative distance of the cell along that line measured so it could be properly placed in a generic LSO. Any cells within the margin of the LSO were designated marginal using the criteria of Helfert and Schwartz (*Helfert and Schwartz, 1986*; *1987*).

## Statistics

Data analysis and statistical tests were done using custom scripts written in MATLAB. Statistical analyses of population data were done using statistical tests as mentioned in Results. For parametric tests data distribution was assumed to be normal but this was not formally tested. Exact p values are given. Statistical significance was defined as p<0.05.

## Supplementary text
### Axon collaterals

It should be emphasized that axon collaterals are often very difficult to distinguish so many of the other illustrated cells may have also displayed collaterals that were missed. No local collaterals in LSO were seen arising from principal cell axons, but the class 5, multiplanar and one of the marginal cells had a 'local' collateral which could be seen within the LSO in the immediate vicinity of the labeled cell body. The multiplanar cell local collateral is seen in *Figure 10* (cell 15): the upper camera lucida is the cell body, dendritic tree and initial part of the main axon (a) and the lower camera lucida is the cell body and the local axon collaterals. E.M. of one of the terminals of this collateral was seen synapsing on another cell body within the LSO with a location and synaptic coverage that indicated it was a marginal cell.

Perhaps most surprising was that 3 of the labeled cells gave off collaterals that headed to the ipsi-lateral lateral nucleus of the trapezoid body (LNTB; one of these cells has been reported before (*Franken et al., 2016*). Two of these cells were principal cells (cells labeled 2 and 5 in *Figure 10*) and one was the multiplanar cell. All 3 of these cells had main axons that gave off this collateral and then headed into the ipsilateral lateral lemniscus. *Figure 10A* shows the LNTB collateral of one of the principal cells (1) within the LNTB and illustrates some of the fairly large swellings seen on such collaterals, presumed to be synaptic terminals (white arrow). One of the LNTB collaterals of another principal cell (labeled 5) was embedded in plastic so it was possible to verify this. *Figure 10B/C* illustrate two of these swellings on the LNTB collateral of the principal cell (2). One (*Figure 10B*, white

asterisk) synapsed on a large dendrite (d) while the other (*Figure 10C*, white asterisk) synapsed on an LNTB cell body (c.b.) immediately adjacent to a large unlabeled terminal (red outline, black asterisk). That large unlabeled terminal is likely to arise from a globular bushy cell axon that we (*Smith et al., 1991*; *Franken et al., 2016*) and others (*Tolbert et al., 1982*; *Spirou et al., 1990*; *Spirou and Berrebi, 1996*) have shown to terminate on LNTB cell bodies. Most LSO cells projecting to the ipsilateral IC are glycinergic while most projecting contralaterally are glutamatergic (*Saint Marie et al., 1989*; *Saint Marie and Baker, 1990*; *Glendenning et al., 1992*) so it is likely that at least one or more of these LSO to LNTB projections is inhibitory.

## Acknowledgements

We thank Anna Thiessen for her help in performing the electron microscopic analysis, Dr. Eric Verschooten for help with recording and analysis software and Dr. Mark Sayles for help with the regularity analysis.

## Additional information

### Funding

| Funder | Grant reference number | Author |
| --- | --- | --- |
| Fonds Wetenschappelijk Onderzoek | Ph.D. fellowship | Tom P Franken |
| National Institute on Deafness and Other Communication Disorders | R01 grant DC006212 | Philip X Joris<br>Philip H Smith |
| Bijzonder Onderzoeksfonds | OT-14-118 | Philip X Joris |
| Fonds Wetenschappelijk Onderzoek | G.0961.11 | Philip X Joris |
| Fonds Wetenschappelijk Onderzoek | G.0A11.13 | Philip X Joris |
| Fonds Wetenschappelijk Onderzoek | G.091214N | Philip X Joris |

The funders had no role in study design, data collection and interpretation, or the decision to submit the work for publication.

### Author contributions

Tom P Franken, Conceptualization, Software, Formal analysis, Funding acquisition, Investigation, Visualization, Methodology, Writing—original draft; Philip X Joris, Conceptualization, Resources, Supervision, Funding acquisition, Project administration, Writing—review and editing; Philip H Smith, Conceptualization, Formal analysis, Supervision, Funding acquisition, Investigation, Visualization, Methodology, Writing—original draft, Project administration, Writing—review and editing

### Author ORCIDs

Tom P Franken (iD) http://orcid.org/0000-0001-7160-5152
Philip X Joris (iD) http://orcid.org/0000-0002-9759-5375

### Ethics

Animal experimentation: This study was performed in strict accordance with the recommendations in the Guide for the Care and Use of Laboratory Animals of the National Institutes of Health. All procedures were approved by the KU Leuven Ethics Committee for Animal Experiments (protocol numbers P155/2008, P123/2010, P167/2012, P123/2013, P005/2014).

### Decision letter and Author response

Decision letter https://doi.org/10.7554/eLife.33854.026

Author response https://doi.org/10.7554/eLife.33854.027

## Additional files

### Supplementary files

• Source code 1. Custom analysis scripts.legend: MATLAB code used for non-trivial analyses.
DOI: https://doi.org/10.7554/eLife.33854.020

• Supplementary file 1. Supplementary Table. Overview of morphological and physiological properties of labeled LSO neurons.
DOI: https://doi.org/10.7554/eLife.33854.021

• Transparent reporting form
DOI: https://doi.org/10.7554/eLife.33854.022

### Data availability

As stated in the Transparent Reporting Form, MATLAB figures with embedded data have been made publicly available on Figshare (https://doi.org/10.6084/m9.figshare.6493409).

The following dataset was generated:

| Author(s) | Year | Dataset title | Dataset URL | Database, license, and accessibility information |
|---|---|---|---|---|
| Tom P Franken, Philip X Joris, Philip H Smith | 2018 | MATLAB figures for the article | https://doi.org/10.6084/m9.figshare.6493409 | Available at figshare under a Creative Commons Attribution 4.0 (CC-BY) license (https://figshare.com). |

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
