## [Decision Letter]

Thank you for submitting your article "Principal cells of the brainstem's interaural sound level detector are temporal differentiators rather than integrators" for consideration by *eLife*. Your article has been reviewed by three peer reviewers, including Catherine Emily Carr as Reviewing Editor and Reviewer #1, and the evaluation has been overseen by Andrew King as the Senior Editor. The following individuals involved in review of your submission has also agreed to reveal their identity: Dan H Sanes (Reviewer #3).

The reviewers have discussed the reviews with one another and the Reviewing Editor has drafted this decision to help you prepare a revised submission.

Summary:

This manuscript approaches an old, once thought to be settled, problem in auditory neuroscience. Principal cells in the lateral superior olive in the auditory brainstem are believed to be the cells that encode the interaural level difference (ILD) cue for sound source localization. These cells had been described as "chopping", or responding with very regular interspike timing in response to sound. This description of principle LSO cells stood for over 4 decades. The present manuscript upends this notion by using modern in vivo whole cell recording methods. The authors were able to thoroughly examine structure and function in populations of LSO neurons, and report that the principal neurons do not chop in response to sound, but rather respond only to sound onset. These findings necessitate consideration of new coding strategies.

Essential revisions:

The main concern raised by reviewers is that the relationship between the key finding, that LSO principal cells have onset responses, and ILD processing, should be more carefully addressed. From the data, it seems that the principal neurons would be inhibited at most physiological ILDs. There are a number of suggestions about how to address this point in the appended reviews. To summarize, you could first determine whether the recorded phasic neurons represent biologically relevant ILDs. Since the sample size is not large, it seems reasonable to display the ILD function of every recorded neuron plotted as a function of absolute discharge rate, and color coded for phasic versus sustained responders. These functions could be quantified by plotting ILD dynamic range and midpoint (ILD at which discharge rate is 50% of max). Once you have these results, you could decide whether the original argument could be supported. If the analysis justifies a model in which phasic LSO neurons encode ILD, then you could integrate previous evidence for an ILD temporal coding mechanism at stimulus onset. Pollak's in vivo study in bat IC (1988) quantified the transformation of intensity to latency in bats and demonstrated how this could produce ILD functions for brief stimuli. The latency results in the current study (Figure 7) are consistent with a synaptic mechanism explored in gerbil LSO brain slices (Sanes, 1990; Figures 11-12; discussed on p. 3504).

The consequences of the strong inhibition would be that LSO principal neurons would not be modulated by the ILDs that gerbils would naturally experience for most frequencies, except the very highest portion of their audibility range, which was not sampled here. From Maki and Furukawa (2005; JASA) "…the maximum ILDs for animal No. 521M across all directions were about 5 dB at 5 kHz, 10 dB at 10 kHz, 20-25 dB at 20 kHz, 30-40 dB at 30 kHz and 25-30 dB above 35 kHz." Based on HRTF measurements in gerbils, the neurons shown in the manuscript would not be modulated by source location except for frequencies greater than 20 kHz. Thus, the suggestion by one reviewer that these 'principal' neurons might be involved in tasks other than localization might need to be discussed more in the manuscript. For example, they could be involved in onset ITD detection and/or envelope ITD detection.

If the current dataset does not support a role for phasic neurons in ILD coding, there are two general alternatives. It is possible that phasic cells do encode biologically relevant ILDs, but something about the experimental design prevented the authors from observing it. For example, the stimulus paradigm used to assess ILD coding (constant ipsilateral level while varying contralateral level) may not have been ideal since average level was uncontrolled. Another possibility is that there might be something about the experimental preparation (e.g., anesthesia, opening the bullae, etc.) that might cause the phasic-type response?

Title: You may wish to make your title more broadly accessible, but this could wait until there is a decision on your manuscript.

*Reviewer #1:*

This paper has used in vivo whole cell patch recording and labeling to show that previous studies of the brainstem lateral superior olivary nucleus (LSO) have largely omitted responses from the principal cells. In retrospect this is understandable; in adults these "fast" neurons do not have overshooting action potentials, and thus their spikes are hard to isolate.

This significant finding shows that LSO neuron fire at stimulus onset, and do not "chop" to the stimulus. This changes how we think about the LSO and how it encodes the interaural loudness differences used for sound localization. These new results move the field forward significantly. They also depend on significant technical expertise.

It would be useful if the authors could enlarge on Figure 1B, which shows ILD functions, with the trace in red being the highest frequency recorded. These ILD functions show sensitivity steeply distributed around -20dB; please discuss where these ILDs fall within the gerbil's range.

The distribution of recorded LSO neurons only extends to 10kHz, while gerbil are assumed to have sensitivity to a much wider frequency range. Thus, the author's sample is not comprehensive. Its' likely that the higher frequency arm of the LSO is similar, but the nature of the dataset could be emphasized.

*Reviewer #2:*

This manuscript approaches an old, once thought to be settled, problem in auditory neuroscience. Principal cells comprising the lateral superior olive (LSO) in the auditory brainstem are believed to be the cells that encode the interuaral level difference (ILD) cue to horizontal sound source location. Based on decades-old work, LSO principle cells have been described in terms of their extracellular physiological response characteristics to monaural and binaural sounds. First, in response to monaural sounds at the characteristic frequency (the frequency the cell is 'tuned' to) LSO principle cells respond with a systematic 'chopping' response whereby action potentials are elicited with very regular interspike timing. Second, sound presented to the ear contralateral to the LSO cell are inhibitory, thus giving LSO principle cells their characteristic sensitivity to the ILD cue. This description of principle LSO cells has stood the test of time for over 4 decades. The present manuscript upends this notion. Using modern in vivo whole cell recording methods, the authors were able to more thoroughly examine structure and function in populations of LSO neurons. Here they report that the principle neurons are not in fact the cells that have chopping responses but rather respond only to the onsets of sounds. These results are transformative not only for auditory neuroscience, but by shining a spotlight on possible sources of bias in such studies, there is likely wider appeal to all of systems neuroscience.

The study appears to be carefully conducted and builds on methods that the groups has developed, used and published before. The results support the stated hypotheses. Amazingly I have little to comment on. The paper is well written. I have just a few questions.

First, was there any observation of ipsilateral inhibition? I am thinking of the classic study of Brownell, Manis and Ritz (1979) in unanesthetized cat that showed clear ipsilateral inhibition in vivo. Moreover, the in vitro work of Banks and also Wu.

The notion that the primary neurons in the LSO might act more akin to medial superior olive (MSO) neurons in terms of temporal processing is interesting. There are some recent modeling reports from Go Ashida (PLoS Computational Biology) that are entirely consistent with the type of LSO neurons describe here. An interesting aspect of the Ashida LSO model is that ILD sensitivity is essentially an emergent property of the exquisite temporal sensitivity, which is consistent with the observation in the manuscript where the timing of excitation and inhibition seems to be the reason for ILD sensitivity in the principal cells. Very neat.

That LSO principle cells act as temporal differentiators also fits with several observations regarding the temporal processing of sound by neurons comprising the LSO afferents. In particular the globular bushy cells and the medial nucleus of the trapezoid body (MNTB), the limb of the circuit that provides inhibition to LSO, exhibit exquisite temporal precision to not only low-frequency sounds (Tollin and Yin, 2005; J Neuroscience) but also the low frequency envelopes of high frequency amplitude modulated sounds (Joris LSO AM epapers). These well-timed inputs are more consistent functionally with the temporal differentiator principle neurons describe here rather than the 'classic' temporal integrators of old.

Finally, I think that it would be of value to the field to quantify the rate-ILD functions (eg Figure 1B) so that these characteristics can be compared to the classic literature. For example, for each neuron what was the dynamic ILD range, half maximum rate ILD, and rate-ILD slope. The examples plotted in Figure 1B look much steeper and located a more negative half maximal ILDs than neurons in the literature. Inhibition appears to be much stronger here than in prior papers.

*Reviewer #3:*

The data and analyses from this study demonstrate that a population of lateral superior olivary neurons display transient responses to tones. This type of study is extremely demanding, and is rarely encountered for any region of the nervous system. In this case, the quality of the data is particularly high, and I have only praise for the data collected and the sophisticated, thorough analyses. The authors conclude that the most common type of LSO neuron (principal cell morphology with onset responses) has been markedly under-sampled in all previous in vivo electrophysiology studies, leading to a misunderstanding about the role of this nucleus in binaural processing. The paper makes a good case for this conclusion by arguing that the small action potential amplitude of onset cells likely caused investigators to miss or dismiss this type of LSO neuron. As discussed below, my primary reservations are: (1) whether the percentage of onset cells can be established by the current findings, and (2) whether onset cells encode azimuthal position or some other sound attribute (which would tend to sustain the role of LSO discharge rate in ILD encoding).

Some characteristics of the data set raise concerns about the degree of support for the authors' major interpretation. First, the sample draws from a relatively narrow range of the mediolateral axis. Most of the low frequency lateral limb (<2kHz) and the high frequency medial limb (>10kHz) are unsampled. A second issue relates to the anatomical criteria and how much emphasis should be placed on structure-function relationships. LSO dendritic morphology varies with tonotopic position (J Comp Neurol 294:443), including the hilus where axon bundles appear to orient dendritic branches and the lateral limb where dendrites are more spread out. While the Helfert and Schwartz (1987) paper is strong on many grounds, it must be acknowledged that their study did not present adequate reconstructions of any cell type (their Figure 2). Put another way, it is impossible to know what the variance was for cells characterized as principal or type 5 in the Helfert study. Therefore, I am less confident about using the Helfert paper as a critical basis for the current set of interpretations. A related issue is that the present study has a lower percentage of principal plus class 5 cells (the latter are apparently only distinguishable at the EM level as having fewer somatic synapses), as compared to Helfert. This could be due either to the necessary limits in sample size for this type of study, but there could also be discrepancies in categorization. For neurons categorized as principal and/or class 5, four of ten apparently have no PSTH, one has a large action potential, and one has sustained activity. What are the authors thoughts on these issues?

Another issue arises in regard to the ILD functions which appear to be markedly shifted towards the ipsilateral side (Figure 1B), in comparison to many of the previous reports cited by the authors. The dynamic range of all but one neuron is devoted to level differences more negative than -20 dB which would be largely outside of the natural range. What is the difference between ILD processing by onset versus sustained neurons in terms of azimuthal range? What is the cell type associated with each of the plotted ILD functions (perhaps a color could be used to identify cells with onset responses)? What do these functions look like when plotted against firing rate instead of normalized firing rate? This might clarify the difference between neurons with onset spikes, onset followed by low level sustained firing, and primary or chopper responses. If very few of the recorded LSO neurons can represent biologically relevant azimuthal sound locations based on level difference, then the implication of these findings for the role of LSO in sound localization (or other percepts) is unclear. Sound localization appears to require an intact superior olive (J Neurophysiol 47:987; J Neurophysiol 67:1643; Behav Neurosci 112:432) even though IC neuron ILD functions are robust to superior olivary lesions and can be created de novo (Hear Res 61:73; Brain Res 572:5; J Neurosci 13:2050). Whether or not LSO onset cells form the major LSO cell type, do they have a plausible role in ILD perception?

The above issues illustrate uncertainties about the relationship between the current data set and the many previous reports on LSO physiology, especially whether the findings support the proposed revision to LSO function. The unambiguous, new finding is that LSO contains a significant number of neurons that respond phasically, and these cells project to the inferior colliculus. Less certain is the assertion that onset neurons are the major LSO cell type that represents ILD, and the inference that all previous studies characterized a minor fraction of LSO cells and reached the errant conclusion that ILD processing (in LSO) is primarily encoded by firing rate. It would seem appropriate to test the authors' prediction with a sufficiently powered extracellular study that samples from the full tonotopic axis, and that makes a point of assessing the percentage of onset responses and their relationship to the representation of ILD or some other stimulus feature.

[Editors' note: further revisions were requested prior to acceptance, as described below.]

Thank you for resubmitting your work entitled "Principal cells of the brainstem's interaural sound level detector are temporal differentiators rather than integrators" for further consideration at *eLife*. Your revised article has been favorably evaluated by Andrew King (Senior Editor) and three reviewers, one of whom is a member of our Board of Reviewing Editors.

The manuscript has been improved but there is a small but important point that should be addressed before acceptance, as outlined below, and found in more detail in the response from your Reviewer 3. We think you will welcome this suggestion, which increases integration with auditory psychophysics.

" one could make a persuasive argument that they have identified one neural mechanism that addresses the resolution-integration paradox (Viemeister and Wakefield, 1991). That is, great temporal resolution requires neurons with fast time constants, whereas great temporal integration requires neurons with slower time constants. V and M envisioned a multiple look model with a fast time constant process that could be integrated downstream (or, as V and W said, "stored in memory") to improve detection or discrimination. "

Reviewer 3 also suggests a more conceptual title. For example, "Level difference is processed by parallel circuits that temporally resolve or integrate synaptic inputs."

*Reviewer #1:*

To recap from the previous review, this paper uses in vivo whole cell patch recording and labeling to show that previous studies of the brainstem lateral superior olivary nucleus (LSO) have largely omitted responses from the principal cells. The findings in this manuscript change how we think about the LSO and how it encodes the interaural loudness differences used for sound localization. It therefore moves the field forward significantly. The results also depend on significant technical expertise.

There were some concerns from the first submission about a ipsilateral bias of the responses, such that many ILDs were on the edge of the physiological range. The authors have satisfied my concerns in this respect.

*Reviewer #2:*

The authors have done an acceptable job of responding to the reviews. I have no further substantive concerns.

*Reviewer #3:*

The revised manuscript addresses the major issues that were raised during the initial review, both with new analyses or figures, and with additional discussion. As a result, the core discovery is now better integrated with the literature on LSO processing. The authors are appropriately cautious about the relationship between their findings and auditory perception, but one could make a persuasive argument that they have identified one neural mechanism that addresses the resolution-integration paradox (Viemeister and Wakefield, 1991). That is, great temporal resolution requires neurons with fast time constants, whereas great temporal integration requires neurons with slower time constants. V and M envisioned a multiple look model with a fast time constant process that could be integrated downstream (or, as V and W said, "stored in memory") to improve detection or discrimination. My sense is that the current study provides a biological basis for the paradox, albeit a somewhat different one than V and W envisioned. LSO onset cells serve as the fast time constant process that provides resolution, while LSO sustained cells serve as the integrator. The last paragraph almost gets there, but remains a bit vague about the purpose ("extracting different cues from the auditory nerve").

The title definitely captures the main message for those interested in the role played by LSO neurons in ILD processing, but I believe that the findings could reach a broader audience with more conceptual title. For example, "Level difference is processed by parallel circuits that temporally resolve or integrate synaptic inputs."

---

## [Author Response]

We thank the reviewers for their time and effort in carefully evaluating our manuscript. Their remarks and suggestions were very valuable to us and have helped us to improve the analysis of the data and discussion of the results in the manuscript.

We agree with the reviewers that more extensive analysis of ILD functions is warranted. We have added a new Figure 9, with the population data of ILD functions (more data than in the former Figure 1B where only ILD functions of 11 labeled cells where shown), and show these as absolute spike rates, separately for the onset and post-onset part of the response. These data confirm that ILD-sensitivity of principal cells is limited to sound onset. In Figure 9C we show the ILD at 50% spike rate and ILD range, as suggested by the reviewers. The data shows that there is no significant difference for these metrics between principal and non-principal cells.

Indeed our sample includes a large portion of functions with slopes at rather negative ILDs. We compare this to published gerbil data from Sanes and Rubel, 1988. We think that the most likely explanation for the bias of ILD-functions towards rather negative ILDs is in our choice of a high ipsilateral level to obtain the ILD function. This observation, that the position of ILD functions is not constant as a function of ipsilateral level but “marches leftward” with increases in the ipsilateral level, is an observation that has been repeatedly made. For example, a classical study by Tsuchitani and Boudreau, 1969 shows that most LSO neurons show a decrease in firing rate if the ILD is kept constant but both ipsi- and contralateral level are increased. Similar observations were made by Park et al., 2004 and Tsai et al., 2010 (full references in manuscript). We have limited data that support this finding that higher ipsilateral sound levels can shift the slope towards more negative ILDs: these are now provided as a supplementary figure (see Figure 9—figure supplement 1). Another possibility that we see has to do, as suggested by the reviewers, with the technique to approach the SOC, which requires opening of the middle ear space (see Materials and methods). We were careful to symmetrize the exposure of the bullas and to keep the middle ears clear, but we cannot exclude asymmetries caused by recording on only one side (e.g. seepage of CSF into the ipsilateral hidden dorsal middle ear cavity) causing an attenuation of the ipsilateral sound level, which would cause a shift of ILD curves towards negative ILDs. Note however that systematic threshold differences were not observed in a study of monaural fibers in the trapezoid body which used a similar recording approach (Wei et al., 2017). We added these statements (re. effect of ipsilateral level, and possibility of acoustic asymmetry due to recording) to the manuscript.

At a more general level, while agreeing that there are many questions left to be answered in the LSO circuit, we are not too concerned about the bias of ILD-functions towards negative ILDs. First, there is no such thing as an absolute physiological range of ILDs. For example, ILDs increase at closer distances of the sound source. The data reported by Maki and Furakawa do not allow to infer any level-dependent changes to the physiological range for ILD. Second, the specific technical restrictions of the experimental technique used, in particular the opening of the bullas, do not make this the best preparation to study the issues raised by the reviewers. A more functionally (rather than mechanistically) oriented study would explore the binaural interaction with stimuli that are more natural in spectrum, timing, and binaural levels (e.g. where the average binaural level is constant, as suggested by one of the reviewers). This is probably best done using a virtual space approach in a preparation with intact middle ears.

We agree with the reviewers that the latency results are consistent with in vitro and in vivo data in the literature, and have added to the Discussion to point this out.

Title: You may wish to make your title more broadly accessible, but this could wait until there is a decision on your ms.

We think the title really captures the main message with a minimal number of words, and is broadly accessible for a scientific audience, but we are open for suggestions for changes.

Reviewer #1:

*This paper has used* in vivo *whole cell patch recording and labeling to show that previous studies of the brainstem lateral superior olivary nucleus (LSO) have largely omitted responses from the principal cells. In retrospect this is understandable; in adults these "fast" neurons do not have overshooting action potentials, and thus their spikes are hard to isolate.*

This significant finding shows that LSO neuron fire at stimulus onset, and do not "chop" to the stimulus. This changes how we think about the LSO and how it encodes the interaural loudness differences used for sound localization. These new results move the field forward significantly. They also depend on significant technical expertise.It would be useful if the authors could enlarge on Figure 1B, which shows ILD functions, with the trace in red being the highest frequency recorded. These ILD functions show sensitivity steeply distributed around -20dB; please discuss where these ILDs fall within the gerbil's range.The distribution of recorded LSO neurons only extends to 10kHz, while gerbil are assumed to have sensitivity to a much wider frequency range. Thus, the author's sample is not comprehensive. Its' likely that the higher frequency arm of the LSO is similar, but the nature of the dataset could be emphasized.

The reason for the restricted range of CFs is due to our approach to the nucleus. In brain slice experiments it is possible to insert the electrode anywhere in the body of a nucleus and to select cells based on their location relative to the boundaries of the LSO. In our in vivo experiments we made a ventral-to- dorsal approach to the LSO with the patch electrodes. This approach allows only a limited range of angles of electrode penetration, because there are several vital structures that cannot be breached (eardrum, venous sinus in medial wall of bulla). This means that our initial contact was always with the ventral margin of the nucleus and increases the chances that our first encounter will be with a non-principal marginal cell. After recording from and potentially labeling a cell the electrode is typically withdrawn and the rest of the nucleus not penetrated. As can be seen from our Figure 2 showing labeled cells, roughly 1 out of 4 of our cells were marginal (because that’s what we hit first) compared to Helfert and Schwartz who reported that roughly 1 out of 16 LSO cells were marginal cells.

We have added a paragraph to Discussion to highlight this issue. As reviewer 3 suggests, additional work is necessary to more fully sample along the tonotopic axis, but one has to keep in mind that extracellular recording techniques appear to undersample principal cells. As of today, we do not see a workable technique to obtain an unbiased characterization of the different response types in this nucleus: it will likely require an (as yet unavailable) imaging technique with very high spatial and temporal resolution.

Reviewer #2:

*This manuscript approaches an old, once thought to be settled, problem in auditory neuroscience. Principal cells comprising the lateral superior olive (LSO) in the auditory brainstem are believed to be the cells that encode the interuaral level difference (ILD) cue to horizontal sound source location. Based on decades-old work, LSO principle cells have been described in terms of their extracellular physiological response characteristics to monaural and binaural sounds. First, in response to monaural sounds at the characteristic frequency (the frequency the cell is 'tuned' to) LSO principle cells respond with a systematic 'chopping' response whereby action potentials are elicited with very regular interspike timing. Second, sound presented to the ear contralateral to the LSO cell are inhibitory, thus giving LSO principle cells their characteristic sensitivity to the ILD cue. This description of principle LSO cells has stood the test of time for over 4 decades. The present manuscript upends this notion. Using modern* in vivo whole cell recording methods, the authors were able to more thoroughly examine structure and function in populations of LSO neurons. Here they report that the principle neurons are not in fact the cells that have chopping responses but rather respond only to the onsets of sounds. These results are transformative not only for auditory neuroscience, but by shining a spotlight on possible sources of bias in such studies, there is likely wider appeal to all of systems neuroscience.The study appears to be carefully conducted and builds on methods that the groups has developed, used and published before. The results support the stated hypotheses. Amazingly I have little to comment on. The paper is well written. I have just a few questions.

*First, was there any observation of ipsilateral inhibition? I am thinking of the classic study of Brownell, Manis and Ritz (1979) in unanesthetized cat that showed clear ipsilateral inhibition* in vivo*. Moreover, the* in vitro *work of Banks and also Wu.*

We have only obtained a small amount of responses to tones with frequencies different from CF or BF. In a few cells we do see some evidence for ipsilaterally evoked IPSPs, but this is subtle. We added a comment to point this out, as it is a possible contributing factor to the non-monotonic level functions.

The notion that the primary neurons in the LSO might act more akin to medial superior olive (MSO) neurons in terms of temporal processing is interesting. There are some recent modeling reports from Go Ashida (PLoS Computational Biology) that are entirely consistent with the type of LSO neurons describe here. An interesting aspect of the Ashida LSO model is that ILD sensitivity is essentially an emergent property of the exquisite temporal sensitivity, which is consistent with the observation in the manuscript where the timing of excitation and inhibition seems to be the reason for ILD sensitivity in the principal cells. Very neat.

As mentioned above, we now refer to the Ashida et al. study in the Discussion.

That LSO principle cells act as temporal differentiators also fits with several observations regarding the temporal processing of sound by neurons comprising the LSO afferents. In particular the globular bushy cells and the medial nucleus of the trapezoid body (MNTB), the limb of the circuit that provides inhibition to LSO, exhibit exquisite temporal precision to not only low-frequency sounds (Tollin and Yin, 2005; J Neuroscience) but also the low frequency envelopes of high frequency amplitude modulated sounds (Joris LSO AM epapers). These well-timed inputs are more consistent functionally with the temporal differentiator principle neurons describe here rather than the 'classic' temporal integrators of old.Finally, I think that it would be of value to the field to quantify the rate-ILD functions (eg Figure 1B) so that these characteristics can be compared to the classic literature. For example, for each neuron what was the dynamic ILD range, half maximum rate ILD, and rate-ILD slope. The examples plotted in Figure 1B look much steeper and located a more negative half maximal ILDs than neurons in the literature. Inhibition appears to be much stronger here than in prior papers.

We refer to our reply above about the population analysis of ILD sensitivity (Figure 9).

Reviewer #3:

*The data and analyses from this study demonstrate that a population of lateral superior olivary neurons display transient responses to tones. This type of study is extremely demanding, and is rarely encountered for any region of the nervous system. In this case, the quality of the data is particularly high, and I have only praise for the data collected and the sophisticated, thorough analyses. The authors conclude that the most common type of LSO neuron (principal cell morphology with onset responses) has been markedly under-sampled in all previous* in vivo electrophysiology studies, leading to a misunderstanding about the role of this nucleus in binaural processing. The paper makes a good case for this conclusion by arguing that the small action potential amplitude of onset cells likely caused investigators to miss or dismiss this type of LSO neuron. As discussed below, my primary reservations are: (1) whether the percentage of onset cells can be established by the current findings, and (2) whether onset cells encode azimuthal position or some other sound attribute (which would tend to sustain the role of LSO discharge rate in ILD encoding).Some characteristics of the data set raise concerns about the degree of support for the authors' major interpretation. First, the sample draws from a relatively narrow range of the mediolateral axis. Most of the low frequency lateral limb (<2kHz) and the high frequency medial limb (>10kHz) are unsampled.

We have addressed this issue above (also raised by reviewer 1).

A second issue relates to the anatomical criteria and how much emphasis should be placed on structure-function relationships. LSO dendritic morphology varies with tonotopic position (J Comp Neurol 294:443), including the hilus where axon bundles appear to orient dendritic branches and the lateral limb where dendrites are more spread out. While the Helfert and Schwartz (1987) paper is strong on many grounds, it must be acknowledged that their study did not present adequate reconstructions of any cell type (their Figure 2). Put another way, it is impossible to know what the variance was for cells characterized as principal or type 5 in the Helfert study. Therefore, I am less confident about using the Helfert paper as a critical basis for the current set of interpretations.

The reviewer is correct that the Helfert and Schwartz paper has a few problems. LSO cell dendritic trees were reconstructed from Golgi-stained cells in 120 micron sections and as a result much of the tree was often not available. In addition, the Helfert and Schwartz electron microscopic evaluation of different cell types was not done on their Golgi labeled cells, where at least some of the features of the dendritic tree could be used. Instead, they looked at unlabeled cells and based their cell-type classification on the orientation of the dendrites as they came off the cell body and on the shape and location of the cell body. In addition, because the class 5 cells were not distinguishable from principal cells at the light microscopic level and because they did not do an unbiased sampling of principal vs. class 5 cells at the E.M. level it is impossible to know the relative percentages of the 2 cell classes. Qualitatively they simply indicated that the class 5 class constituted a much lower percentage. Unfortunately, our sample is too small to add anything to help resolve this question. Despite these indications that this is not an “ideal” study for comparison with our data, we do feel it is the best available choice. First the data is collected from the same species and second this is the only paper that we are aware of where electron microscopic features are used as criteria for cell subtypes. There is a nice LSO development paper that also evaluates cell types in the rat by Rietzel and Friauf (1998) where they used electrode-labeled LSO cells with more of the dendritic tree intact and did a more quantitative evaluation of LSO cell classes but unfortunately it was from a different species and no electron microscopic evaluation of the cells was made. There is also a nice paper (Sanes et al., 1990) that does a careful quantitative evaluation of dendritic configurations of gerbil LSO cells and their differences at different LSO locations but this paper focused on the principal cell class and no electron microscopy was done. So, it is our feeling that the Helfert and Schwartz paper is the best option we have.

We have added a reference to the study by Rietzel and Friauf to the manuscript.

A related issue is that the present study has a lower percentage of principal plus class 5 cells (the latter are apparently only distinguishable at the EM level as having fewer somatic synapses), as compared to Helfert. This could be due either to the necessary limits in sample size for this type of study, but there could also be discrepancies in categorization. For neurons categorized as principal and/or class 5, four of ten apparently have no PSTH, one has a large action potential, and one has sustained activity. What are the authors thoughts on these issues?

We have addressed the issue of a different distribution of cell types (relatively more marginal cells and relatively less principal cells) in response to the issue raised by reviewer 1 about the CF distribution of our sample (see above).

About the absence of PSTH in the table: for one cell we did not obtain a full PSTH but the obtained responses during frequency tuning show that this cell fired at sound onset. For the remaining three cells, we did not obtain enough data to be able to judge firing to tones at CF or BF.

One labeled principal cell indeed had a relatively large action potential amplitude, but it was smaller than most of the action potential amplitudes of non-principal cells. We note that the error in measuring action potential amplitude in vivo is probably fairly large due to filtering of the signal in the presence of suboptimal access resistance/bridge balance. We added a phrase to results to emphasize this. This would affect especially the measurement of large action potentials (thus non-principal cells) and therefore would make the measured difference between principal and non-principal cells’ action potential amplitudes smaller than it actually is. When we combine action potential amplitude and upper cut-off frequency of spontaneous activity, we do see a clear clustering of the cells in two groups.

The dot rasters show indeed that a low-level sustained activity is possible in some principal cells, but this is nowhere near the sustained activity of chopping non-principal neurons. Analysis of the ILD functions shows that this low-level activity is not meaningful post-sound onset in terms of ILD tuning (new Figure 9, A and B)

*Another issue arises in regard to the ILD functions which appear to be markedly shifted towards the ipsilateral side (Figure 1B), in comparison to many of the previous reports cited by the authors. The dynamic range of all but one neuron is devoted to level differences more negative than -20 dB which would be largely outside of the natural range. What is the difference between ILD processing by onset versus sustained neurons in terms of azimuthal range? What is the cell type associated with each of the plotted ILD functions (perhaps a color could be used to identify cells with onset responses)? What do these functions look like when plotted against firing rate instead of normalized firing rate? This might clarify the difference between neurons with onset spikes, onset followed by low level sustained firing, and primary or chopper responses. If very few of the recorded LSO neurons can represent biologically relevant azimuthal sound locations based on level difference, then the implication of these findings for the role of LSO in sound localization (or other percepts) is unclear. Sound localization appears to require an intact superior olive (J Neurophysiol 47:987; J Neurophysiol 67:1643; Behav Neurosci 112:432) even though IC neuron ILD functions are robust to superior olivary lesions and can be created* de novo *(Hear Res 61:73; Brain Res 572:5; J Neurosci 13:2050). Whether or not LSO onset cells form the major LSO cell type, do they have a plausible role in ILD perception?*

We have addressed these issues above (see reply to Essential Revisions). In short, population analysis of ILD tuning shows that principal neurons are only sensitive to ILD at sound onset, but we do not find other significant differences between ILD functions of principal and non-principal cells, such as ILD at 50% spike rate or ILD range covered by the slope of the function.

The above issues illustrate uncertainties about the relationship between the current data set and the many previous reports on LSO physiology, especially whether the findings support the proposed revision to LSO function. The unambiguous, new finding is that LSO contains a significant number of neurons that respond phasically, and these cells project to the inferior colliculus. Less certain is the assertion that onset neurons are the major LSO cell type that represents ILD, and the inference that all previous studies characterized a minor fraction of LSO cells and reached the errant conclusion that ILD processing (in LSO) is primarily encoded by firing rate. It would seem appropriate to test the authors' prediction with a sufficiently powered extracellular study that samples from the full tonotopic axis, and that makes a point of assessing the percentage of onset responses and their relationship to the representation of ILD or some other stimulus feature.

We agree that there are many remaining uncertainties and that additional work is needed to more fully sample along the tonotopic axis and to evaluate the relative numbers of onset vs. sustained responders. However, our results strongly suggest that “traditional” extracellular recordings will bias against principal cells, so it will not be easy to address these remaining issues. Perhaps (future) imaging techniques (which however would need to have high temporal and spatial resolution) will allow differentiation between physiological cells types and enable a spatial overview. Another possibility, which we are currently exploring, are axonal recordings from the lateral lemniscus (Bremen and Joris, J Neurosci, 2013).

[Editors' note: further revisions were requested prior to acceptance, as described below.]

The manuscript has been improved but there is a small but important point that should be addressed before acceptance, as outlined below, and found in more detail in the response from your reviewer 3. We think you will welcome this suggestion, which increases integration with auditory psychophysics." one could make a persuasive argument that they have identified one neural mechanism that addresses the resolution-integration paradox (Viemeister and Wakefield, 1991). That is, great temporal resolution requires neurons with fast time constants, whereas great temporal integration requires neurons with slower time constants. V and M envisioned a multiple look model with a fast time constant process that could be integrated downstream (or, as V and W said, "stored in memory") to improve detection or discrimination. "Reviewer 3 also suggests a more conceptual title. For example, "Level difference is processed by parallel circuits that temporally resolve or integrate synaptic inputs."

Reviewer #3:

The revised manuscript addresses the major issues that were raised during the initial review, both with new analyses or figures, and with additional discussion. As a result, the core discovery is now better integrated with the literature on LSO processing. The authors are appropriately cautious about the relationship between their findings and auditory perception, but one could make a persuasive argument that they have identified one neural mechanism that addresses the resolution-integration paradox (Viemeister and Wakefield, 1991). That is, great temporal resolution requires neurons with fast time constants, whereas great temporal integration requires neurons with slower time constants. V and M envisioned a multiple look model with a fast time constant process that could be integrated downstream (or, as V and W said, "stored in memory") to improve detection or discrimination. My sense is that the current study provides a biological basis for the paradox, albeit a somewhat different one than V and W envisioned. LSO onset cells serve as the fast time constant process that provides resolution, while LSO sustained cells serve as the integrator. The last paragraph almost gets there, but remains a bit vague about the purpose ("extracting different cues from the auditory nerve").

We particularly like the statement “The authors are appropriately cautious about the relationship between their findings and auditory perception”… The reference to the work of Viemeister and Wakefield (1991) and particularly the references in that work to earlier discussions by Green (1984) and de Boer (1984) indeed provide interesting and relevant discussions of the theme of multiple time scales of integration. We added a couple of sentences to end the Discussion referring to this work. Rather than reference these 3 seminal papers, we reference the review by Viemeister and Plack, 1993. Admittedly, the comment we added is again quite cautious and not as ambitious as the suggestion by the reviewer. However, we also added a reference to recent work by Brown and Tollin, 2016 who recently addressed different time scales of integration in physiology and perception, along the lines suggested by the reviewer.

The title definitely captures the main message for those interested in the role played by LSO neurons in ILD processing, but I believe that the findings could reach a broader audience with more conceptual title. For example, "Level difference is processed by parallel circuits that temporally resolve or integrate synaptic inputs."

We find that our title "Principal cells of the brainstem's interaural sound level detector are temporal differentiators rather than integrators" better captures the essence of our study. The title suggested by the reviewer has a somewhat different conceptual emphasis but we believe the processes of differentiation and integration are conceptually equally important and appealing to draw reader attention.